# KoALA: KL–L0 Adversarial Detector via Label Agreement

## Abstract

Deep neural networks are highly susceptible to adversarial attacks, which pose significant risks to security- and safety-critical applications. We present KoALA (**KL–L0** Adversarial detection via **L**abel **A**greement), a novel, semantics-free adversarial detector that requires no architectural changes or adversarial retraining. KoALA operates on a simple principle: it detects an adversarial attack when class predictions from two complementary similarity metrics disagree. These metrics—KL divergence and an $L_0$-based similarity—are specifically chosen to detect different types of perturbations. The KL divergence metric is sensitive to dense, low-amplitude shifts, while the $L_0$-based similarity is designed for sparse, high-impact changes. We provide a **formal proof of correctness** for our approach. The only training required is a simple fine-tuning step on a pre-trained image encoder using clean images to ensure the embeddings align well with both metrics. This makes KoALA a lightweight, plug-and-play solution for existing models and various data modalities. Our extensive experiments on ResNet/CIFAR-10 and CLIP/Tiny-ImageNet confirm our theoretical claims. When the theorem's conditions are met, KoALA consistently and effectively detects adversarial examples. On the full test sets, KoALA achieves a precision of 0.94 and a recall of 0.81 on ResNet/CIFAR-10, and a precision of 0.66 and a recall of 0.85 on CLIP/Tiny-ImageNet.

## 1 Introduction

The increasing deployment of machine learning and deep learning models in safety-critical applications—such as autonomous driving, medical imaging, and security—underscores the need for robust and reliable systems. However, neural networks remain vulnerable to adversarial attacks, where small, often imperceptible perturbations to an input can cause the model to make a confident misclassification (Biggio et al., 2013; Xiao et al., 2018a;b; Szegedy et al., 2013). Protecting these models from such manipulation is a critical security and safety concern.

Defenses against adversarial attacks generally fall into three categories (Aldahdooh et al., 2022). The first, *verification and certification*, aims to formally prove model robustness within a defined perturbation set (Khedr & Shoukry, 2024; Liu et al., 2021). While these methods provide strong guarantees, they do not actively improve the model's behavior in deployment. The second, *proactive defenses*, such as adversarial training and randomized smoothing, harden models by retraining or modifying their architecture (Madry et al., 2017b; Cohen et al., 2019; Shafahi et al., 2019). These methods can be computationally expensive, often require prior knowledge of attack types, and may lag behind novel attack strategies. The final category, *reactive detection*, augments a deployed model with a separate detector to flag adversarial inputs without altering the core network.

We focus on this reactive detection paradigm. Prior work in this area has largely pursued two main avenues. The first involves *add-on detectors*, which rely on empirical observations of adversarial examples, such as their intrinsic statistics or the effects of feature space (Xu et al., 2018; Ma & Liu, 2019; Ma et al., 2018; Meng & Chen, 2017). Other methods train a separate detector head using adversarial examples (Metzen et al., 2017; Grosse et al., 2017). While these methods can be effective, they typically *lack formal guarantees of correctness*. The second involves *semantics-driven detectors* that leverage external information, such as label text, auxiliary classifiers, or handcrafted cues (Zhang et al., 2023; Zhou et al., 2024; Muller et al., 2024). While powerful, these approaches depend on

domain-specific priors that may not always be available or vary across different deployments and data modalities. Critically, they also *lack proof of correctness* for their detection conditions.

To address this issue, we present a novel perspective based on the geometry of norm-bounded adversarial perturbations. As shown in Figure 1, we observe that energy-bounded attacks manifest either as (i) *dense, low-amplitude* shifts across many coordinates or (ii) *sparse, high-impact* shifts on few coordinates. These characteristics are naturally captured by two complementary similarity measures: KL divergence, which is sensitive to broad, small-magnitude output shifts; and an $L_0$-based score, which is sensitive to sparse, large-magnitude coordinate changes.

In this work, we propose KOALA, a light-weight and semantics-free adversarial detector that flags input as attack when predictions derived from our two complementary metrics—KL divergence and the $L_0$-based score—disagree. The only required training is a brief fine-tuning of an image encoder to align embeddings with both metrics simultaneously. This makes KOALA a simple, plug-and-play solution for existing models without the need for adversarial training or architectural changes.

Our approach is distinguished by a *formal mathematical guarantee*. We prove that under norm-bounded perturbations and mild assumptions on the separation between class prototypes and the input embedding, each metric induces a distinct prediction stability band. Once the margins between the classes are sufficiently large, no single perturbation can keep the input within both bands simultaneously. This mutual exclusivity forces a disagreement between the two metrics, leading to guaranteed detection. Our extensive experiments on ResNet/CIFAR-10 and CLIP/Tiny-ImageNet corroborate this theory, demonstrating robust attack identification without relying on semantic priors, architectural modifications, or costly adversarial retraining.

Our core contributions are summarized as follows:

- We introduce KOALA, a novel, plug-in adversarial detector based on the disagreement between KL divergence and $L_0$-based predictions.
- We provide a theoretical proof of correctness that defines the explicit conditions under which this disagreement—and thus detection—is guaranteed to occur.
- We propose a lightweight training recipe that only requires fine-tuning an encoder with clean images, avoiding the need for architectural changes or adversarial examples.
- Our comprehensive experimental results demonstrate strong detection performance, aligning with our theory and offering a valuable complement to existing robust training and certification methods.

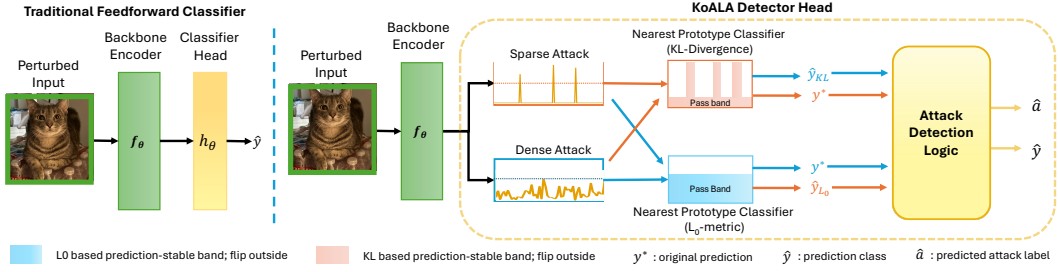

Figure 1: **Motivation for combining KL and $L_0$ as an attack detector.** With an energy bound adversarial input, $\|\boldsymbol{\delta}\|_2 \le \epsilon$, the resulting perturbation may be *dense* (distributed) or *sparse* (concentrated). Each metric defines a prediction-stability band: inside the band the label remains $y^*$; outside it flips to $\hat{y}$. Dense attacks typically violate the $L_0$ band (green), while sparse attacks violate the KL band (orange). When two classification decisions disagree, we can detect adversarial attacks.

## 2 RELATED WORK

**Detectors trained with adversarial examples.** An intuitive way of train an adversarial detector is to train it on generated adversarial examples (Metzen et al., 2017; Grosse et al., 2017; Lee et al., 2024). While effective against the attacks seen during training, these detectors typically rely on prior knowledge of the threat model and can degrade under newly crafted or adaptive attacks. Our work is orthogonal to theirs in that our approach does not require adversarial training.

**Detectors utilizing intrinsic statistics of attacks.** Compared to clean samples, adversarial inputs often exhibit systematic statistical deviations designed to fool neural networks. Leveraging this observation, prior work distinguishes clean from adversarial inputs by extract residual and structural information of clean data (Kong et al., 2025) or probing regularities in feature or activation space, e.g., invariant checking over internal activations (NIC) (Ma & Liu, 2019), prediction inconsistency under input transformations (feature squeezing) (Xu et al., 2018), local intrinsic dimensionality (LID) statistics (Ma et al., 2018), autoencoder-based reformers/detectors (MagNet) (Meng & Chen, 2017), Mahalanobis (Lee et al., 2018), CADet (Guille-Escuret et al., 2023), Bayesian-based uncertainty (Feinman et al., 2017), class-disentanglement (Yang et al., 2021) and adversarial direction comparision (Hu et al., 2019). These methods are generally empirical and lack formal *proof-of-correctness* guarantees against adaptive adversaries. While we provide explicit theoretical conditions under which our detector is provably correct, specifying when adversarial examples must be detected.

**Semantics- and knowledge-driven detection.** Attacks can also be detected by examining semantic inconsistencies at inference using domain knowledge and reasoning modules(Mumuni & Mumuni, 2024), e.g., MLN/GCN pipelines for certifiable robustness (Zhang et al., 2023), knowledge-enabled graph detection (Zhou et al., 2024; Song et al., 2025), and part-level reasoning for object tracking defenses (VOGUES) (Muller et al., 2024). These approaches can be powerful but have limitations across modalities and tasks, as their effectiveness depends on semantics and specific domain knowledge. In contrast, our method is semantics-free: it operates purely on representation geometry via a $KL/L_0$ disagreement criterion and provides detector-specific correctness conditions, yielding a lightweight, plug-in detector for safety-critical models.

## 3 METHODOLOGY

### 3.1 THE KOALA DETECTOR AND KOALA HEAD

We consider a neural network classifier comprised of two main components: i) a backbone encoder $f_\theta : \mathcal{I} \to \mathbb{R}^d$ that maps input from the data space $\mathcal{I}$ (e.g., images) to feature embedding $\in \mathbb{R}^d$; ii) a classifier head $h_\theta : \mathbb{R}^d \to \{1, \ldots, m\}$ uses the embedding to determine the final class.

In a traditional feedforward neural network, the backbone encoder corresponds to all layers up to the penultimate layer, while the classifier head is the final output layer (e.g., a fully connected layer followed by a softmax). Our method, KOALA, replaces this conventional classifier head with a novel component, which we term the KOALA Detector, operates on the embeddings produced by the backbone encoder to simultaneously classify the input and flag it as an attack when necessary.

As shown in Figure 2, the KOALA Detector operates as a nearest prototype classifier (Snell et al., 2017), which determines the predicted class $\hat{y} \in \{1, \ldots, m\}$ by finding the prototype vector—the pre-computed centroid for each class—that is closest to the input's feature embedding in the normalized feature space, i.e., for feature vector $\boldsymbol{p} = f_\theta(\boldsymbol{I})$ of input $\boldsymbol{I}$, the nearest prototype classifier head

$$\hat{y} = \arg\min_k \textbf{Distance}(\boldsymbol{c}_k, \boldsymbol{p})$$

for some **Distance** function and pre-selected prototype vectors (also known as class centroids) $\boldsymbol{c}_1, \ldots, \boldsymbol{c}_m$. This effectively classifies input based on its proximity to representatives of each class.

Traditional nearest prototype classifiers use a single distance metric(e.g., Euclidean) to find the closest class prototype. In contrast, KOALA is designed to leverage multiple, complementary metrics for classification and adversarial detection. The motivation behind KOALA is the observation that adversarial perturbations can manifest in two distinct ways under an energy-limited budget:

- *Sparse, High-Impact Perturbations:* Few feature dimensions are modified with a large magnitude.
- *Dense, Low-Amplitude Perturbations:* Many feature dimensions are modified by small magnitude.

These two types of attacks are difficult to detect with a single metric. KOALA addresses this by using a combination of $L_0$ and $KL$ divergence metrics:

- $KL$ Divergence: This metric measures the shift in the output probability distribution. It is particularly sensitive to dense, low-amplitude perturbations that subtly influence the model's overall

output, even if no single feature dimension is drastically altered. The $KL$ Divergence is defined as:

$$KL(\boldsymbol{c}\|\boldsymbol{p}) = \sum_{i=1}^{d} c_i \log \frac{c_i}{p_i}. \tag{1}$$

- $L_0$ distance: This metric measures the number of dimensions in the feature vector that have been perturbed above a certain threshold. It is therefore highly sensitive to sparse, high-impact changes, making it effective at detecting targeted, "surgical" attacks. The $L_0$ distance metric is defined as:

$$L_0(\boldsymbol{c}, \boldsymbol{p}) = \mathbf{card}\Big(\{i : |c_i - p_i| - \tau \cdot \mu(\boldsymbol{c}, \boldsymbol{p}) > 0\}\Big), \tag{2}$$

where $\mathbf{card}(\{.\})$ denotes the cardinality of the set, $\mu(\boldsymbol{c}, \boldsymbol{p}) = \frac{1}{d} \sum_{i=1}^{d} |c_i - p_i|$ is the average distance across all the entries of $|\boldsymbol{c} - \boldsymbol{p}|$, and $\tau \in [0, 1]$ is a threshold parameter. In other words, the $L_0$ metric counts the number of features whose value are above a certain threshold relative to the average value of the feature vector.

The KOALA Detector operates by simultaneously leveraging the two complementary metrics above. For a given input embedding $\boldsymbol{p}$, the detector computes both the $KL$-divergence and the $L_0$-based distance to all class prototype vectors $\boldsymbol{c_k}$. These computations yield two distinct class predictions:

$$\hat{y}_{\mathrm{KL}} = \arg\min_k KL(\boldsymbol{c_k}, \boldsymbol{p}), \qquad \hat{y}_{L_0} = \arg\min_k L_0(\boldsymbol{c_k}, \boldsymbol{p}). \tag{3}$$

The core of our detection mechanism lies in the disagreement between these two predictions. An input is declared attacked when the class predicted by the KL-divergence, $\hat{y}_{KL}$, does not match the class predicted by the $L_0$-based metric, $\hat{y}_{L_0}$. In this case, the detector abstains from making a final classification. If the two predictions agree, the input is considered benign, and the shared class prediction becomes the final output. This behavior is formally defined by the following decision rule:

$$(\hat{a}, \hat{y}) = (1, \bot) \text{ if } \hat{y}_{L_0} \neq \hat{y}_{\mathrm{KL}}, \text{ else } (0, \hat{y}_{\mathrm{KL}}). \tag{4}$$

where $\hat{a} \in \{0, 1\}$ is the predicted attack label, with $\hat{a} = 1$ indicating an attack and $\hat{y}$ the final predicted class, with $\bot$ signifying an abstention (no class).

### 3.2 THEORETICAL GUARANTEES

Our proposed method, KOALA, is not merely an empirical defense; it is grounded in a formal mathematical guarantee. We provide a proof of correctness under a set of mild and practical assumptions. The core idea is to show that a single adversarial perturbation cannot simultaneously fool both the KL- and $L_0$-based classifiers.

The following assumptions underpin our main theorem:

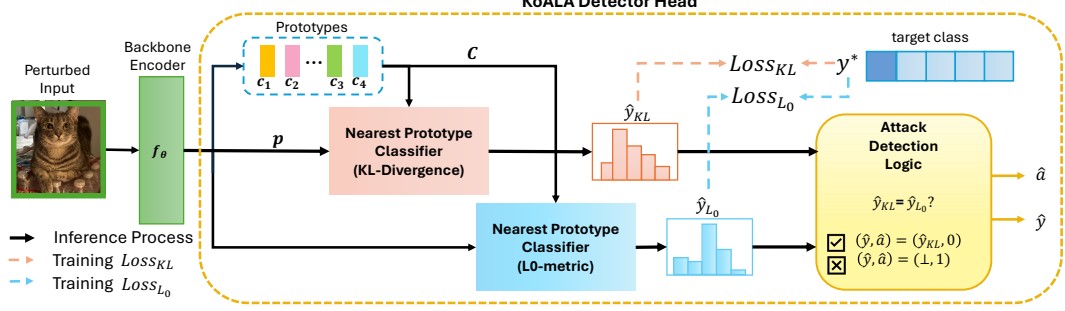

Figure 2: **Training phase:** Class centroids $\boldsymbol{C}$ are computed as the centroid of image embeddings within each class. Each image embedding $\boldsymbol{p}$ is compared with $\boldsymbol{C}$ to compute the $Loss_{KL}$ and $Loss_{L_0}$. The model is trained to make the L0 and KL distances small for the correct class while large for incorrect classes. **Inference phase:** An input image embedding $p$ is compared with class centroids $\boldsymbol{C}$ to calculate $KL$ and $L0$-based predictions $\hat{y}_{KL}$ and $\hat{y}_{L_0}$. The predicted class is accepted only if both metrics agree; otherwise, the system flags the input as an adversarial attack detected ($\hat{a} = 1$).

**A1** *Normalized Feature vector space:* All feature embeddings $f_\theta(\boldsymbol{I})$ and class prototypes $\boldsymbol{c_1}, \ldots, \boldsymbol{c_m}$ are normalized, i.e., their coordinates sum to 1 and are strictly positive. This is satisfied by using a softmax or similar normalization on the feature vectors.

**A2** *Bounded Perturbation:* The adversarial perturbation $\delta$ in the feature space has a limited energy budget, i.e., $\|\delta\| \le \epsilon$. This is a standard assumption in adversarial robustness, following from the Lipschitz continuity of the backbone encoder.

**A3** *Coordinate-wise Bound:* The magnitude of the perturbation on any single coordinate is bounded relative to the original value, $|\delta_i| \le \frac{3}{2}|p_i^*|$. This is a mild and practical condition, as extremely large, coordinate-wise perturbations are rarely effective or imperceptible.

**A4** *Clean Example Alignment:* On clean, unperturbed inputs, both the KL and $L_0$ metrics agree on the true class. This alignment is encouraged by our lightweight fine-tuning procedure, which shapes the embeddings to be meaningful under both metrics.

Building on these assumptions, our central result is Theorem 1, which establishes that a sufficiently large separation between class prototypes guarantees the detection of adversarial attacks.

**Theorem 1.** *If Assumptions A1-A4 are satisfied, and there exists a coordinate $i$ where the gap between the true class prototype $c_i^*$ and the predicted adversarial class prototype $\hat{c}_i$ is sufficiently large (i.e., $|c_i^* - \hat{c}_i| > \Gamma_i(\epsilon)$, for some threshold $\Gamma_i(\epsilon)$), then no perturbation $\delta$ with $\|\delta\| \le \epsilon$ can simultaneously cause both the KL- and $L_0$-based predictions to favor the adversarial class.*

In essence, the theorem proves that the KL and $L_0$ stability bands are mutually exclusive for adversarial perturbations. An attack can push an embedding out of one stability band, causing a prediction flip, but it cannot simultaneously push it out of both. This forces a disagreement, leading to guaranteed detection. This result provides a rigorous foundation for KOALA's effectiveness, showing that if the feature space is properly structured (which our fine-tuning encourages), detection is not a probabilistic outcome but a mathematical certainty.

**Proof Sketch for Theorem 1:** A complete proof of Theorem 1 is provided in the appendix B. Below, we provide a high-level sketch to convey the core intuition behind our guarantee. The proof's central idea is to show that, under a limited energy budget, an adversarial perturbation cannot simultaneously satisfy the conditions required to fool both the KL- and $L_0$-based classifiers. We establish this through three key propositions:

(i) *Necessary Conditions for successful attack on KL-Divergence metric* (Prop. 2): To change the KL-based prediction from the true class prototype $\boldsymbol{c}^*$ to an adversarial class prototype $\hat{\boldsymbol{c}}$, the adversarial perturbation $\delta$ must have a positive inner product with the vector $\hat{\boldsymbol{c}} - \boldsymbol{c}^*$. This condition, means the perturbation must "align" with a particular direction in the feature space.

(ii) *Necessary Conditions for successful attack on $L_0$-metric* Prop. 3): To change the $L_0$ based prediction, the perturbation must alter a minimum number of feature dimensions ($k$) by a significant amount. This consumes a portion of the total perturbation energy ($\|\delta\|$) allowed by the budget. The more dimensions that need to be flipped, the more energy is consumed, and the less is left for other purposes.

(iii) *The Incompatibility Condition* (Prop. 4): We show that these two conditions are fundamentally incompatible. For any given adversarial perturbation, we can always find a threshold $\tau$ for the $L_0$ metric that forces a trade-off. The energy required to satisfy the $L_0$ flip condition (moving a sufficient number of coordinates by a large enough magnitude) leaves insufficient residual energy to satisfy the KL-flip condition (aligning the perturbation with the vector $\hat{\boldsymbol{c}} - \boldsymbol{c}^*$).

(iv) *Conclusion:* The final step proves that such a threshold $\tau$ always exists as long as there is a sufficiently large "coordinate gap" between the true class prototype and the adversarial class prototype. This means that if the feature space is well-structured–which our fine-tuning encourages–no single adversarial perturbation can successfully flip both predictions, forcing them to disagree and enabling our detection mechanism.

### 3.3 Fine-Tuning for Prototype Alignment

Our formal guarantees in Theorem 1 rely on the assumption that on clean inputs, the feature embeddings are well-aligned with their respective class prototypes under both KL-divergence and $L_0$-based metrics (Assumption **A4**). To achieve this, we introduce a lightweight fine-tuning procedure for the backbone encoder $f_\theta$. This procedure is designed to simultaneously minimize the distance

between a clean image embedding and its corresponding class prototype across both metrics, thereby encouraging the "coordinate gap" crucial for our detection method.

Our training objective is a composite loss that penalizes the dissimilarity between image embeddings and their class prototypes. To ensure stable optimization, we first map the KL and $L_0$ distances to a comparable, differentiable, and range-bounded similarity score.

• **KL-similarity loss**: We define the KL-based similarity between a class prototype $c$ and an image embedding $p$ as:

$$\text{sim}_{KL}(\mathbf{c}, \mathbf{p}) = \exp\big(-\text{KL}(\mathbf{c}\|\mathbf{p})\big) \in (0, 1]$$

Using this similarity, we train the encoder with a standard binary cross-entropy loss over a set of positive and negative image-prototype pairs. This loss encourages the similarity of positive pairs (matching image and prototype) to be high and that of negative pairs (mismatched image and prototype) to be low. Formally, we finetune the model using the following loss function:

$$\mathcal{L}_{KL} = -\mathbb{E}_{(i,j)\in\mathcal{P}}[y_{ij}^* \log s_{ij} + (1 - y_{ij}^*) \log(1 - s_{ij})], \quad \text{where } s_{ij} = \text{sim}_{KL}(\mathbf{c}_i, \mathbf{p}_j) \quad (5)$$

Here, $\mathcal{P}$ denotes the set of image-prototype pairs, and $y_{ij}^*$ is a binary label (1 for a matching pair, 0 otherwise).

• $L_0$**-similarity loss**: The $L_0$ distance, which counts the number of perturbed dimensions, is non-differentiable. To make it trainable, we use a smooth, differentiable surrogate. We approximate the $L_0$ metric with a smoothed surrogate function $\widehat{L_0}(\mathbf{c}, \mathbf{p})$ using the sigmoid function to obtain a continuous value. The $L_0$-based similarity is then defined as a normalized, inverse measure of this surrogate:

$$\text{sim}_{L_0}(\mathbf{c}, \mathbf{p}) = 1 - \frac{\widehat{L_0}(\mathbf{c}, \mathbf{p})}{d} \in [0, 1], \quad \text{where } \widehat{L_0}(\mathbf{c}, \mathbf{p}) = \sum_{i=1}^{d} \sigma\left(\frac{|c_i - p_i| - \tau \cdot \mu(\mathbf{c}, \mathbf{p})}{\phi}\right)$$

where $\phi > 0$ is a smoothness parameter and $\sigma(x) = \frac{1}{1+e^{-x}}$ is the sigmoid function. Similar to the KL loss, we use the binary cross entropy loss for $L_0$-based similarity:

$$\mathcal{L}_{L_0} = -\mathbb{E}_{(i,j)\in\mathcal{P}}[y_{ij}^* \log s_{ij} + (1 - y_{ij}^*) \log(1 - s_{ij})], \quad \text{where } s_{ij} = \text{sim}_{L_0}(\mathbf{c}_i, \mathbf{p}_j).$$

• **Total Objective:** The final training objective is a weighted sum of the two similarity losses:

$$\mathcal{L}_{\text{total}} = \omega_{L_0} \mathcal{L}_{L_0} + \omega_{KL} \mathcal{L}_{KL}, \quad (6)$$

where $\omega_{L_0}$ and $\omega_{KL}$ are non-negative mixing weights. This composite loss guides the encoder to produce embeddings that are simultaneously cohesive under both a dense-shift-sensitive metric (KL) and a sparse-shift-sensitive metric $L_0$, which is a key requirement for KOALA's guaranteed detection.

# 4 EXPERIMENTS

Our experiments evaluate KOALA's performance on two distinct architectures and datasets, employing standard adversarial attacks to test its effectiveness.

## 4.1 EXPERIMENTAL SETUP

• **Models and Datasets**. We use two models to demonstrate KOALA's versatility: a ResNet-18 model on CIFAR-10 and a CLIP model on Tiny-ImageNet. For both datasets, we randomly split the development sets into two equal halves to serve as the test and validation sets.

- **ResNet-18 on CIFAR-10:** We start with a baseline ResNet-18 backbone trained on CIFAR-10 (Krizhevsky et al., 2009). The final fully connected layer (classifier head) is removed to produce image embeddings. Class prototypes (centroids) $c_1, \ldots, c_m$ are computed as the mean embedding of all training examples for each class. The backbone is finetuned using the composite loss described in the Fine-Tuning section, with SGD optimizer, learning rate $1 \times 10^{-3}$, weight decay $5 \times 10^{-4}$, momentum 0.9, and batch size 128. The loss weights are set to $\omega_{L_0} = 0.9$ and $\omega_{KL} = 0.1$ (as $L_0$ is harder to optimize) and the hyperparameters are $\tau = 0.75$ and $\phi = 0.5$.

- **CLIP on Tiny-ImageNet:** We also fine-tune the pre-trained CLIP ViT-B/32 model on the Tiny-ImageNet dataset. The class prototypes $c^k$ here are obtained by using the CLIP text encoder with prompt "a photo of [CLASS]". SGD is used for fine-tuning with learning rate $1 \times 10^{-4}$, weight decay 0, momentum 0.9, and batch size 128. The loss weights again $\omega_{L_0} = 0.9$ and $\omega_{\mathrm{KL}} = 0.1$.

- **Adversarial Attacks:** We generate a variety of adversarial examples using established attack methods. We report results on clean accuracy, adversarial accuracy, and adversarial detection rate. All attacks are constrained by the $\ell_\infty$ norm with $\epsilon \in \{2/255, 4/255\}$ and a batch size of 128.

  - **PGD (Projected Gradient Descent) (Madry et al., 2017a):** A classic iterative attack used to generate adversarial examples for both the ResNet and CLIP models.

  - **CW (Carlini & Wagner, 2017) Attack:** A powerful, optimization-based attack on both models.

  - **AutoAttack (Croce & Hein, 2020):** A suite of four diverse attacks used to reliably test robustness, serving as a robust benchmark against both models.

### 4.2 EXPERIMENT 1: VERIFYING THEORETICAL GUARANTEES

- **Experiment Objective:** We validate our central theorem by evaluating KOALA's performance on examples that either satisfy or do not satisfy the conditions of Theorem 1. The primary goal is to show that when the conditions are met, attack detection is guaranteed. We partition the test sets of both CIFAR-10 and Tiny-ImageNet into two groups (i) **Theorem-Compliant Samples:** Inputs that satisfy the conditions of Theorem 1, specifically the sufficient inter-class prototype separation and (ii) **Non-Compliant Samples:** Inputs that do not satisfy these conditions. Table 1 provides a breakdown of the number of samples (sample size columns) in each group for both datasets, highlighting that the ResNet model on CIFAR-10 exhibits a larger inter-class separation than the CLIP model on Tiny-ImageNet. This is likely due to the massive scale of CLIP's pre-training data, which can lead to a more compact, less-separable embedding space for a smaller, specialized task like Tiny-ImageNet.

- **Evaluation Metrics:** To evaluate detection performance, we define a confusion matrix where an "attacked" input (i.e., $a = 1$) is considered a positive result as follows:

$$\mathrm{TP} := \left[a = 1\right] \wedge \left[(\widehat{a}, \widehat{y}) = (1, \bot) \ \vee \ (\widehat{a}, \widehat{y}) = (0, y^*)\right], \qquad \mathrm{TN} := \left[a = 0\right] \wedge \left[(\widehat{a}, \widehat{y}) = (0, y^*)\right],$$

$$\mathrm{FP} := \left[a = 0\right] \wedge \left[(\widehat{a}, \widehat{y}) = (1, \bot) \ \vee \ (\widehat{a}, \widehat{y}) = (0, \neg y^*)\right], \quad \mathrm{FN} := \left[a = 1\right] \wedge \left[(\widehat{a}, \widehat{y}) = (0, \neg y^*)\right].$$

Using these definitions, we report standard metrics: Accuracy, Precision, Recall, and F1-score: $\mathrm{Acc} = \frac{TP+TN}{N}$, $\mathrm{Prec} = \frac{TP}{TP+FP}$, $\mathrm{Rec} = \frac{TP}{TP+FN}$, $\mathrm{F1} = \frac{2\,\mathrm{Prec}\,\mathrm{Rec}}{\mathrm{Prec}+\mathrm{Rec}}$, $N = TP+TN+FP+FN$.

- **Results and Analysis:** Table 1 summarizes the overall performance. Notably, the recall scores are all 1.0 on the Theorem-compliant subset. This means every adversarial attacked input that satisfies the theorem's conditions is successfully detected, providing strong empirical support for our theoretical guarantee. The Accuracy and precision for theorem-compliant examples are 1.0 as well. This is because the theory assumes that clean, compliant examples are correctly classified by both the KL and $L_0$ heads, leading to prediction agreement and preventing false alarms.

As our theory predicts, the Theorem-compliant subset achieves a substantially higher Precision and Recall compared to the non-compliant subset, confirming that when the inter-class prototype separation is sufficiently large, adversarial perturbations are forced to cause a disagreement between the KL and $L_0$ heads, leading to more reliable attack detection.

### 4.3 EXPERIMENT 2: ABLATION STUDY ON METRIC COMBINATIONS

- **Experiment Objective:** We run an ablation study to validate our choice of using KL-divergence and $L_0$-based metrics for attack detection. We compare the performance of our proposed KL+$L_0$ combination against other plausible metric pairings: $L_0$+Cosine, KL+Cosine, and $L_0$+KL+Cosine. For each combination, we fine-tune the backbone encoder using a composite loss tailored to the specific metrics, then evaluate the detector's performance. It's important to note that all models were fine-tuned exclusively with clean, non-adversarial images. No adversarial training was performed.

- **Results and Analysis:** The results, summarized in Table 2, show that the KL+$L_0$ combination consistently yields the best performance on the ResNet/CIFAR-10 setup, achieving the highest scores

| Model | Attack Perturbation | Thm. 1 Compliant Samples | | | | Non Compliant Samples | | | | |
| --- | --- | --- | --- | --- | --- | --- | --- | --- | --- | --- |
| | | Sample Size | Acc | Prec | Rec | F1 | Sample Size | Acc | Prec | Rec | F1 |
| ResNet-CIFAR-10 | $\ell_\infty^{2/255}$ | 3345 | **1.0** | **1.0** | **1.0** | **1.0** | 1655 | 0.63 | 0.73 | 0.42 | 0.53 |
| | $\ell_\infty^{4/255}$ | 2967 | **1.0** | **1.0** | **1.0** | **1.0** | 2033 | 0.66 | 0.78 | 0.45 | 0.57 |
| CLIP-TinyImageNet | $\ell_\infty^{2/255}$ | 510 | **1.0** | **1.0** | **1.0** | **1.0** | 4490 | 0.67 | 0.63 | 0.84 | 0.72 |
| | $\ell_\infty^{4/255}$ | 556 | **1.0** | **1.0** | **1.0** | **1.0** | 4444 | 0.65 | 0.62 | 0.80 | 0.70 |

Table 1: Results from Experiment 1: Detector metrics—accuracy, precision, recall, and F1—for ResNet-18 and CLIP (ViT-B/32) backbone finetuned with our $\mathcal{L}_{\text{total}}$ objective in equation 6 and evaluated under PGD on the two subsets: images that *satisfy* Theorem 1 vs. those that *do not*.

across all four key metrics: Accuracy, Precision, Recall, and F1-score. This confirms our hypothesis that KL-divergence and $L_0$-based metrics are highly complementary. The KL metric effectively captures dense, distribution-level shifts that often go undetected by other measures, while the $L_0$ metric is uniquely sensitive to sparse, high-impact changes. Their combined use allows the detector to identify a wider range of adversarial attack types.

The results on the CLIP/Tiny-ImageNet setup, however, show that the $L_0$+KL+Cosine combination slightly outperforms the others. This unexpected finding is an interesting artifact of the model's behavior. As shown in Table 6, the model fine-tuned with the $L_0$+KL+Cosine loss exhibits a very low adversarial accuracy. This indicates that a single adversarial perturbation pushes the embedding into a region where all three metrics are essentially "randomly guessing" a class. The probability of all three classifiers independently guessing the same incorrect class is extremely low, leading to frequent disagreements and, consequently, a high attack detection rate.

This outcome on the CLIP model underscores a critical distinction: a high detection rate does not always equate to a truly robust model. While the $L_0$+KL+Cosine setup appears effective at flagging attacks on CLIP, it does so by breaking the underlying classification, rather than by preserving it. This contrasts with the ResNet results, where our KL+$L_0$ combination shows a more balanced approach to robust classification and detection.

| Metric Combinations | Attack Perturbation | ResNet-CIFAR-10 | | | | CLIP-TinyImageNet | | | |
| --- | --- | --- | --- | --- | --- | --- | --- | --- | --- |
| | | Accuracy | Precision | Recall | F1 | Accuracy | Precision | Recall | F1 |
| KL+L0 | $\ell_\infty^{2/255}$ | **0.88** | **0.94** | **0.81** | **0.87** | 0.71 | 0.66 | 0.85 | 0.74 |
| | $\ell_\infty^{4/255}$ | **0.87** | **0.94** | **0.78** | **0.85** | 0.69 | 0.65 | 0.82 | 0.73 |
| L0+Cosine | $\ell_\infty^{2/255}$ | 0.73 | 0.91 | 0.52 | 0.66 | 0.70 | 0.66 | 0.85 | 0.74 |
| | $\ell_\infty^{4/255}$ | 0.68 | 0.89 | 0.41 | 0.56 | 0.68 | 0.64 | 0.79 | 0.71 |
| KL+Cosine | $\ell_\infty^{2/255}$ | 0.78 | 0.92 | 0.62 | 0.74 | 0.70 | 0.66 | 0.82 | 0.73 |
| | $\ell_\infty^{4/255}$ | 0.76 | 0.91 | 0.59 | 0.71 | 0.71 | 0.67 | 0.84 | 0.74 |
| KL+L0+Cosine | $\ell_\infty^{2/255}$ | 0.75 | 0.91 | 0.55 | 0.69 | **0.75** | **0.68** | **0.94** | **0.79** |
| | $\ell_\infty^{4/255}$ | 0.69 | 0.89 | 0.44 | 0.59 | **0.74** | **0.68** | **0.93** | **0.78** |

Table 2: Results from Experiment 2: Comparison of key detector performance metrics (accuracy, precision, recall, F1) for ResNet-18 and CLIP (ViT-B/32) models.

## 4.4 Experiment 3: Overall Adversarial Resilience Across Metric Combinations

• **Experiment Objective:** This experiment moves beyond attack detection metrics to evaluate the overall classification robustness of models fine-tuned with different metric combinations. We report both clean accuracy (performance on benign images) and adversarial accuracy (performance on successfully attacked images that were not detected) to assess how each fine-tuning objective impacts the underlying model's resilience. Again, our fine-tuning procedure is intentionally lightweight, relying solely on clean images. Unlike traditional adversarial defenses, our approach does not require costly adversarial examples or specialized training routines

• **Results and Analysis for ResNet Model on CIFAR-10:** We fine-tuned a ResNet-18 backbone using seven different objectives: Cosine similarity, $L_0$, KL, $L_0$+KL, Cosine+KL, Cosine+$L_0$, and Cosine+KL+$L_0$. The results in Table 3 show that all models maintain comparable clean accuracy, indicating that the fine-tuning process does not degrade the model's core classification ability.

However, the models yield starkly different adversarial accuracies. Our proposed KL+$L_0$ objective achieves the strongest adversarial performance because KL-divergence and $L_0$-based metrics are fundamentally complementary: KL excels at capturing dense, distribution-level shifts, while $L_0$ is sensitive to sparse, high-impact changes. Optimizing both simultaneously forces the embeddings to be robust against a wider variety of adversarial perturbations, leading to better overall resilience.

In contrast, any objective that includes the Cosine similarity leads to significantly lower adversarial robustness. The Cosine similarity encourages an angular alignment that conflicts with the the per-dimension alignment of KL and $L_0$. The resulting optimization trade-off degrades the model's ability to resist attacks, highlighting why simply adding more metrics is not always beneficial.

| Models | Image Encoder | Clean Image Accuracy (%) | PGD attack(%) | | CW attack(%) | | Auto attack(%) | |
|---|---|---|---|---|---|---|---|---|
| | | | $\ell_\infty^{2/255}$ | $\ell_\infty^{4/255}$ | $\ell_\infty^{2/255}$ | $\ell_\infty^{4/255}$ | $\ell_\infty^{2/255}$ | $\ell_\infty^{4/255}$ |
| Baseline model | ResNet18 | **95.16** | 45.5 | 33.11 | 45.99 | 35.98 | 45.49 | 31.95 |
| **Note:** All finetuning was done using clean images only | Cosine Similarity | 94.98 | 45.8 | 37.8 | 37.80 | 33.00 | 35.40 | 22.02 |
| | KL | 89.50 | 41.48 | 29.00 | 39.06 | 30.78 | 40.74 | 30.62 |
| | $L_0$ | 94.96 | 49.08 | 32.66 | 47.02 | 35.30 | 42.56 | 35.88 |
| | KL+$L_0$ | 94.78 | **57.32** | **54.60** | **57.52** | **54.08** | **52.28** | **51.12** |
| | Cosine+$L_0$ | 94.76 | 43.98 | 32.22 | 44.78 | 36.18 | 44.94 | 35.92 |
| | KL+Cosine | 94.36 | 55.60 | 51.32 | 45.02 | 34.08 | 45.48 | 34.18 |
| | KL+$L_0$+Cosine | 94.48 | 44.66 | 32.86 | 45.42 | 34.52 | 45.84 | 35.52 |

Table 3: Clean and adversarial accuracy for the ResNet-18 backbone fine-tuned with seven different single/composite embedding objectives under a PGD attack. The KL+$L_0$ objective demonstrates superior adversarial accuracy, highlighting the complementary nature of these two metrics.

| Models | Image Encoder | Clean Image Accuracy (%) | PGD attack | | Auto attack | | CW attack | |
|---|---|---|---|---|---|---|---|---|
| | | | $\ell_\infty^{2/255}$ | $\ell_\infty^{4/255}$ | $\ell_\infty^{2/255}$ | $\ell_\infty^{4/255}$ | $\ell_\infty^{2/255}$ | $\ell_\infty^{4/255}$ |
| baseline model | CLIP(ViT-B/32) | 57.88 | 0.38 | 0.28 | 0.01 | 0.01 | 0.0 | 0.0 |
| **Note:** All finetuning was done using clean images only | Cosine Similarity | **62.44** | 33.74 | 33.72 | 3.22 | 0.07 | 3.06 | 0.05 |
| | $L_0$ | 54.34 | 53.31 | 43.42 | **25.43** | **18.35** | **37.49** | **13.67** |
| | KL | 57.65 | **60.02** | **58.87** | 19.35 | 11.76 | 25.69 | 11.16 |
| | KL+$L_0$ | 55.88 | 26.50 | 25.47 | 16.18 | 9.57 | 11.91 | 5.84 |
| | Cosine+$L_0$ | 56.46 | 16.28 | 16.09 | 1.03 | 0.02 | 1.15 | 0.01 |
| | Cosine+KL | 57.62 | 55.01 | 53.87 | 5.25 | 0.44 | 5.02 | 0.39 |
| | KL+$L_0$+Cosine | 56.30 | 14.93 | 14.72 | 0.97 | 0.06 | 1.14 | 0.01 |

Table 4: Clean and adversarial accuracy for the CLIP ViT-B/32 backbone fine-tuned with seven different single/composite embedding objectives under a PGD attack. The KL+$L_0$ objective demonstrates superior adversarial accuracy, highlighting the complementary nature of these two metrics.

• **Results and Analysis for CLIP Model on Tiny-ImageNet:** Table 4 presents the results for the fine-tuned CLIP model. Unlike the ResNet, the $L_0$-only fine-tuning objective yields the highest adversarial robustness, which can be attributed to the models' different training histories and architectures.

The CLIP model is pre-trained on a massive dataset using a cosine-contrastive objective, which naturally encourages inter-class variation to be concentrated in a few principal directions of the high-dimensional text embedding space. Because of this pre-existing sparsity-aware structure, enforcing further alignment via the $L_0$-based metric is especially effective. Conversely, the ResNet model is trained from scratch on a smaller dataset (CIFAR-10) using a cross-entropy loss, which encourages class separations that are dispersed over many coordinates. For such a model, a single metric is insufficient. The combined KL+$L_0$ criterion becomes necessary to simultaneously account for both dense and sparse perturbations, thereby realizing the necessary gains in adversarial robustness.

## 5 ETHICS STATEMENT:

Adversarial attacks pose significant risks to the safety and security of machine learning systems, particularly in sensitive applications such as autonomous vehicles and medical diagnostics. Our work on the KOALA's detection method aims to mitigate these risks by providing a robust, theoretically grounded defense. We believe that by enhancing the security of deep neural networks, our research contributes positively to the ethical deployment of AI technology. This work does not use any sensitive personal data or create new privacy risks. It focuses on improving model robustness against malicious manipulation, thereby helping to ensure that AI systems operate as intended and can be trusted in real-world, safety-critical scenarios. We are committed to transparency and will make our code and models publicly available to facilitate further research and independent verification.

## 6 REPRODUCIBILITY STATEMENT:

We provide all details needed to reproduce our results. Section 3 specifies our KOALA's architecture, theoretical guarantees, and training objectives; Section 4 describes training/evaluation datasets, architectures, attack settings, hyperparameters, and evaluation metrics. The appendix provides full proof of our theorem. We also provide an anonymous repository in the supplementary materials with training/evaluation scripts.

## 7 USAGE OF LLM

We used the large language model (LLM) as a general-purpose writing assistant for copy-editing (grammar, phrasing, and concision) and LaTeX formatting suggestions. The LLM did not generate ideas, claims, proofs, figures, or results. All technical content and experiments were authored and verified by the authors, who take full responsibility for the paper. LLMs are not authors.

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

# A  Confusion Counts of Experiment 1

In Experiment 1, we validated Theorem 1 by testing KOALA on inputs that either satisfy or violate the conditions of Theorem 1. We split CIFAR-10 and Tiny-ImageNet into (i) **Theorem-Compliant** (sufficient inter-class prototype separation) and (ii) **Non-Compliant** subsets. Here in Table 5 we report the number of samples (sample size columns) in each group for both datasets and the raw data of the confusion counts (TP/TN/FP/FN) which were used to calculate the accuracy, precision, recall, and F1 in Table 1.

| Model | Attack | Thm. 1 Compliant Samples | | | | | Non Compliant Samples | | | | |
|---|---|---|---|---|---|---|---|---|---|---|---|
| $(KL + L_0)$ | Perturbation | Sample Size | TP | FN | FP | TN | Sample Size | TP | FN | FP | TN |
| ResNet-CIFAR-10 | $\ell_\infty^{2/255}$ | 3345 | 3345 | 0 | 0 | 3345 | 1655 | 690 | 965 | 260 | 1395 |
| | $\ell_\infty^{4/255}$ | 2967 | 2967 | 0 | 0 | 2967 | 2033 | 919 | 1114 | 260 | 1773 |
| CLIP-TinyImageNet | $\ell_\infty^{2/255}$ | 510 | 510 | 0 | 0 | 510 | 4490 | 3762 | 728 | 2206 | 2284 |
| | $\ell_\infty^{4/255}$ | 556 | 556 | 0 | 0 | 556 | 4444 | 3555 | 889 | 2206 | 2238 |

Table 5: Experiment 1 Raw Results on (a) CIFAR-10 (ResNet-18) and (b) Tiny-ImageNet (CLIP ViT-B/32) show the number of test images (sample size) that **satisfy** or do **not satisfy** the conditions of Thm. 1. The table also shows the **Confusion metrics**—TP,TN,FP,FN —for both backbones finetuned with our $\mathcal{L}_{\text{total}}$ objective in equation 6 and evaluated under PGD on the two subsets.

# B  Proof of Theorem 1

## B.1  Necessary condition for successful attack on $KL$ detector

**Proposition 2** (Necessary condition for successful attack on KL detector). *Let $\boldsymbol{p}^* \in \mathbb{R}^d$ be the input embedding (feature vector) of the clean image and $\hat{\boldsymbol{p}} \in \mathbb{R}^d$ be the input embedding of the adversarially attacked image, i.e., $\hat{\boldsymbol{p}} = \boldsymbol{p}^* + \boldsymbol{\delta}$ where $\boldsymbol{\delta}$ is the adversarial perturbation of the input embedding. Similarly, let $\boldsymbol{c}^* \in \mathbb{R}^d$ be the prototype vector (or class centroid) of the target class and $\hat{\boldsymbol{c}} = \hat{\boldsymbol{c}}_{\hat{y}_{KL}} \in \mathbb{R}^d$ be the prototype vector (or class centroid) of the predicted class $\hat{y}_{KL} \in \{1, \ldots, m\}$ based on the $KL$ distance. Consider a successful attack on the KL detector (i.e., $\hat{\boldsymbol{c}} \neq \boldsymbol{c}^*$) and assume the Assumptions A1-A4 are satisfied, then the following inequality holds:*

$$\sum_{i=1}^{d} \left(\hat{c}_i - c_i^*\right) \frac{\delta_i}{p_i^*} \ > \ \Delta KL(\boldsymbol{p}^*),$$

*where:*

$$\Delta KL(\boldsymbol{p}^*) = KL(\hat{\boldsymbol{c}}||\boldsymbol{p}^*) - KL(\boldsymbol{c}^*||\boldsymbol{p}^*), \quad and \quad \Delta KL(\hat{\boldsymbol{p}}) = KL(\hat{\boldsymbol{c}}||\hat{\boldsymbol{p}}) - KL(\boldsymbol{c}^*||\hat{\boldsymbol{p}}). \quad (7)$$

*Proof.* First note that, since $\boldsymbol{p}^*$ is the input embedding of the clean image and $\boldsymbol{c}^*$ is its corresponding target class centroid, then $\boldsymbol{c}^*$ is the closest class centroid to $\boldsymbol{p}^*$ (Assumption A4) and hence $\Delta KL(\boldsymbol{p}^*) > 0$. Similarly, it follows from the definition of $\Delta KL(\hat{\boldsymbol{p}})$ and the assumption that the attack is successful (i.e., $\hat{\boldsymbol{p}}$ is predicted as $\hat{\boldsymbol{c}}$ class), that $\Delta KL(\hat{\boldsymbol{p}}) \leq 0$.

Substituting in the previous equations yield:

$$\Delta KL(\boldsymbol{p}^*) = KL(\hat{\boldsymbol{c}}||\boldsymbol{p}^*) - KL(\boldsymbol{c}^*||\boldsymbol{p}^*)$$

$$= \sum_{i=1}^{d} (\hat{c}_i log(\hat{c}_i) - \hat{c}_i log(p_i^*)) - \sum_{i=1}^{d} (c_i^* log(c_i^*) - c_i^* log(p_i^*))$$

$$= \sum_{i=1}^{d} (\hat{c}_i log(\hat{c}_i) - c_i^* log(c_i^*)) - \sum_{i=1}^{d} (\hat{c}_i log(p_i^*) - c_i^* log(p_i^*)) > 0, \quad (8)$$

and:

$$\Delta KL(\hat{\boldsymbol{p}}) = KL(\hat{\boldsymbol{c}}||\hat{\boldsymbol{p}}) - KL(\boldsymbol{c}^*||\hat{\boldsymbol{p}})$$

$$= \sum_{i=1}^{d}(\hat{c}_i log(\hat{c}_i) - \hat{c}_i log(\hat{p}_i)) - \sum_{i=1}^{d}(c_i^* log(c_i^*) - c_i^* log(\hat{p}_i))$$

$$= \sum_{i=1}^{d}(\hat{c}_i log(\hat{c}_i) - c_i^* log(c_i^*)) - \sum_{i=1}^{d}(\hat{c}_i log(\hat{p}_i) - c_i^* log(\hat{p}_i)) < 0. \quad (9)$$

Subtracting Equation 8-Equation 9, we get:

$$\sum_{i=1}^{d}(\hat{c}_i log(\hat{p}_i) - c_i^* log(\hat{p}_i)) - \sum_{i=1}^{d}(\hat{c}_i log(p_i^*) - c_i^* log(p_i^*)) = \sum_{i=1}^{d}(\hat{c}_i - c_i^*)(log(\hat{p}_i) - log(p_i^*))$$

$$= \Delta KL(\boldsymbol{p}^*) - \Delta KL(\hat{\boldsymbol{p}})$$

$$> \Delta KL(\boldsymbol{p}^*). \quad (10)$$

Expanding the left hand side of the inequality using Taylor Expansion of $log(\hat{p}')$ yields:

$$log(\hat{\boldsymbol{p}}) = log(\boldsymbol{p}^* + \boldsymbol{\delta}) = log(\boldsymbol{p}^*) + \boldsymbol{\delta}^T \nabla_{\boldsymbol{p}^*} log(\boldsymbol{p}^*) + \frac{1}{2}\boldsymbol{\delta}^T \nabla_{\boldsymbol{p}^*}^2 log(\boldsymbol{p}^*)\boldsymbol{\delta} + \mathcal{R}_3$$

$$= log(\boldsymbol{p}^*) + \frac{\boldsymbol{\delta}^T}{\boldsymbol{p}^*} - \boldsymbol{\delta}^T diag(\frac{1}{2(\boldsymbol{p}^*)^2})\boldsymbol{\delta} + \mathcal{R}_3. \quad (11)$$

Based on Taylor's remainder theorem, the error of truncating after the 2nd order is bounded. The remainder term in the Taylor expansion is:

$$\mathcal{R}_3 = \frac{\nabla_{\boldsymbol{p}^*}^3 log(\boldsymbol{p}^* + \theta\boldsymbol{\delta})}{6}\boldsymbol{\delta}^3, \quad \text{for some } \theta \in [0, 1].$$

Since the third derivative of $log(p^* + \theta\delta)$ is $\nabla_{\boldsymbol{p}^*}^3 log(\boldsymbol{p}^* + \theta\boldsymbol{\delta}) = \frac{2}{(\boldsymbol{p}^*+\theta\boldsymbol{\delta})^3}$, we conclude:

$$|\mathcal{R}_3| = \frac{|\nabla_{\boldsymbol{p}^*}^3 log(\boldsymbol{p}^* + \theta\boldsymbol{\delta})|}{6}|\boldsymbol{\delta}|^3 = \sum_i \frac{|\delta_i|^3}{3|p_i^* + \theta\delta_i|^3}, \quad (12)$$

which leads to:

$$-\boldsymbol{\delta}^T diag(\frac{1}{2(\boldsymbol{p}^*)^2})\boldsymbol{\delta} + \mathcal{R}_3 \leq -\boldsymbol{\delta}^T diag(\frac{1}{2(\boldsymbol{p}^*)^2})\boldsymbol{\delta} + |\mathcal{R}_3|$$

$$\leq -\boldsymbol{\delta}^T diag(\frac{1}{2(\boldsymbol{p}^*)^2})\boldsymbol{\delta} + \sum_i \frac{|\delta_i|^3}{3|p_i^* + \theta\delta_i|^3}$$

$$= -\sum_i \frac{|\delta_i|^2}{2|p_i^*|^2} + \sum_i \frac{|\delta_i|^3}{3|p_i^* + \theta\delta_i|^3}. \quad (13)$$

Since $\theta$, $p^*$, and $\delta$ all lie in the interval $[0, 1]$ (thanks to Assumption A1), we observe that the term:

$$\sum_i \frac{|\delta_i|^3}{3|p_i^* + \theta\delta_i|^3},$$

increases as $\theta$ decreases. In particular, when $\theta = 0$, it reaches its maximum value of $\sum_i \frac{|\delta_i|^3}{3|p_i^*|^3}$. Hence, we can rewrite this as:

$$\sum_i \frac{|\delta_i|^3}{3|p_i^*|^3} = \sum_i \left(\frac{|\delta_i|^2}{2|p_i^*|^2} \cdot \frac{2|\delta_i|}{3|p_i^*|}\right).$$

Under the assumption that $|\delta_i| < \frac{3}{2}|p_i^*|$ for all dimensions $i$ (Assumption A3), we have:

$$\frac{2|\delta_i|}{3|p_i^*|} \leq 1 \quad \Rightarrow \quad \frac{|\delta_i|^3}{3|p_i^* + \theta\delta_i|^3} \leq \frac{|\delta_i|^3}{3|p_i^*|^3} \leq \frac{|\delta_i|^2}{2|p_i^*|^2}.$$

Thus, we can bound the remainder term $\mathcal{R}_3$ in the Taylor expansion as:

$$-\boldsymbol{\delta}^\top \operatorname{diag}\left(\frac{1}{2(\boldsymbol{p}^*)^2}\right)\boldsymbol{\delta} + \mathcal{R}_3 \le -\sum_i \frac{|\delta_i|^2}{2|p_i^*|^2} + \sum_i \frac{|\delta_i|^3}{3|p_i^* + \theta\delta_i|^3} < 0.$$

By combining above Taylor expansion with Equation 10:

$$\Delta KL(\boldsymbol{p}^*) < (\hat{\boldsymbol{c}} - \boldsymbol{c}^*)(log(\hat{\boldsymbol{p}}) - log(\hat{\boldsymbol{p}}^*))$$

$$= (\hat{\boldsymbol{c}} - \boldsymbol{c}^*)(\frac{\boldsymbol{\delta}^T}{\boldsymbol{p}^*} - \boldsymbol{\delta}^T diag(\frac{1}{2(\boldsymbol{p}^*)^2})\boldsymbol{\delta} + \mathcal{R}_3)$$

$$< (\hat{\boldsymbol{c}} - \boldsymbol{c}^*)\frac{\boldsymbol{\delta}^T}{\boldsymbol{p}^*}. \tag{14}$$

$\square$

## B.2 NECESSARY CONDITION FOR SUCCESSFUL ATTACK ON $L_0$ DETECTOR

**Proposition 3** (Necessary condition for successful attack on $L_0$ detector)**.** *Let $\boldsymbol{p}^* \in \mathbb{R}^d$ be the input embedding (feature vector) of the clean image and $\hat{\boldsymbol{p}} \in \mathbb{R}^d$ be the input embedding of the adversarially attacked image, i.e., $\hat{\boldsymbol{p}} = \boldsymbol{p}^* + \boldsymbol{\delta}$ where $\boldsymbol{\delta}$ is the adversarial perturbation of the input embedding. Similarly, let $\boldsymbol{c}^* \in \mathbb{R}^d$ be the prototype vector (or class centroid) of the target class and $\hat{\boldsymbol{c}} = \hat{\boldsymbol{c}}_{\hat{y}_{L_0}} \in \mathbb{R}^d$ be the prototype vector (or class centroid) of the predicted class $\hat{y}_{L_0} \in \{1, \dots, m\}$ based on the $L_0$ distance. Consider a successful attack on the $L_0$ detector (i.e., $\hat{\boldsymbol{c}} \ne \boldsymbol{c}^*$) and assume the Assumptions A1-A4 are satisfied, then there exists a nonempty set of indices $\mathbb{S} \subseteq \{1, \dots, d\}$, where for each $i \in \mathbb{S}$ the following holds:*

$$|\delta_i| \ge min\left\{ ||\hat{c}_i - p_i^*| \ - \ \tau\mu(\hat{\boldsymbol{c}}, \boldsymbol{p}^*)| - \frac{\tau||\boldsymbol{\delta}||_1}{d}, ||c_i^* - p_i^*| \ - \ \tau\mu(\boldsymbol{c}^*, \boldsymbol{p}^*)| - \frac{\tau||\boldsymbol{\delta}||_1}{d} \right\}, \tag{15}$$

*while for all other indices $\delta_j \notin \mathbb{S}$, the following holds:*

$$|\delta_j| \le \sqrt{\epsilon^2 - k\left[min\left\{ ||\hat{c}_i - p_i^*| - \tau\mu(\hat{\boldsymbol{c}}, \boldsymbol{p}^*)| - \frac{\tau||\boldsymbol{\delta}||_1}{d}, ||c_i^* - p_i^*| - \tau\mu(\boldsymbol{c}^*, \boldsymbol{p}^*)| - \frac{\tau||\boldsymbol{\delta}||_1}{d} \right\}\right]^2}.$$

*where $k$ is the cardinality of the set $\mathbb{S}$, i.e., $k = |\mathbb{S}|$. Moreover, the cardinality $k$ satisfies $k = |\mathbb{S}| \ge \Delta L_0(\boldsymbol{p}^*)$ where:*

$$\Delta L_0(\boldsymbol{p}^*) = L_0(\hat{\boldsymbol{c}}, \boldsymbol{p}^*) - L_0(\boldsymbol{c}^*, \boldsymbol{p}^*), \quad and \quad \Delta L_0(\hat{\boldsymbol{p}}) = L_0(\hat{\boldsymbol{c}}, \hat{\boldsymbol{p}}) - L_0(\boldsymbol{c}^*, \hat{\boldsymbol{p}}).$$

*Proof.* First note that, since $\boldsymbol{p}^*$ is the embedding of the clean input and $\boldsymbol{c}^*$ is its corresponding target class centroid, then $c^*$ is the closest class to $\boldsymbol{p}^*$ (Assumption A4) and hence $\Delta L_0(\boldsymbol{p}^*) > 0$. Similarly, it follows from the definition of $\Delta L_0(\hat{\boldsymbol{p}})$ and the assumption that the attack is successful (i.e., $\hat{\boldsymbol{p}}$ is predicted as the class whose centroid is $\hat{\boldsymbol{c}}$), that $\Delta L_0(\hat{\boldsymbol{p}}) \le 0$. For sake of presentation, we define the following sets:

$$\mathbb{A} = \{i : |\hat{c}_i - p_i^*| - \tau \cdot \mu(\hat{\boldsymbol{c}}, \boldsymbol{p}^*) > 0, |c_i^* - p_i^*| - \tau \cdot \mu(\boldsymbol{c}^*, \boldsymbol{p}^*) \le 0\} \tag{16}$$

$$\mathbb{B} = \{i : |\hat{c}_i - p_i^*| - \tau \cdot \mu(\hat{\boldsymbol{c}}, \boldsymbol{p}^*) \le 0, |c_i^* - p_i^*| - \tau \cdot \mu(\boldsymbol{c}^*, \boldsymbol{p}^*) > 0\} \tag{17}$$

$$\mathbb{C} = \{i : |\hat{c}_i - p_i^* - \delta_i| - \tau \cdot \mu(\hat{\boldsymbol{c}}, \hat{\boldsymbol{p}}) > 0, |c_i^* - p_i^* - \delta_i| - \tau \cdot \mu(\boldsymbol{c}^*, \hat{\boldsymbol{p}}) \le 0\} \tag{18}$$

$$\mathbb{D} = \{i : |\hat{c}_i - p_i^* - \delta_i| - \tau \cdot \mu(\hat{\boldsymbol{c}}, \hat{\boldsymbol{p}}) \le 0, |c_i^* - p_i^* - \delta_i| - \tau \cdot \mu(\boldsymbol{c}^*, \hat{\boldsymbol{p}}) > 0\}. \tag{19}$$

Using this notation, we can rewrite $\Delta L_0(\boldsymbol{p}^*)$ and $\Delta L_0(\hat{\boldsymbol{p}})$ as:

$$\Delta L_0(\boldsymbol{p}^*) = |\mathbb{A}| - |\mathbb{B}| > 0 \tag{20}$$

$$\Delta L_0(\hat{\boldsymbol{p}}) = |\mathbb{C}| - |\mathbb{D}| \le 0, \tag{21}$$

where $|\mathbb{A}|, |\mathbb{B}|, |\mathbb{C}|$, and $|\mathbb{D}|$ denote the cardinality (i.e., the number of elements) of the corresponding sets.

Subtracting Equation 20-Equation 21 yields:

$$\Delta L_0(\boldsymbol{p}^*) - \Delta L_0(\hat{\boldsymbol{p}}) > \Delta L_0(\boldsymbol{p}^*) \Rightarrow \Delta L_0(\boldsymbol{p}^*) - \Delta L_0(\hat{\boldsymbol{p}}) > 0 \tag{22}$$

$$\Rightarrow |\mathbb{A}| - |\mathbb{B}| - |\mathbb{C}| + |\mathbb{D}| > 0 \tag{23}$$

$$\Rightarrow (|\mathbb{A}| - |\mathbb{C}|) + (|\mathbb{D}| - |\mathbb{B}|) > 0. \tag{24}$$

This implies that at least one of the terms $|\mathbb{A}| - |\mathbb{C}|$ or $|\mathbb{D}| - |\mathbb{B}|$ must be positive, since the sum of two quantities is positive only if at least one of them is positive. We proceed with case analysis.

• **Case 1:** If $|\mathbb{A}| - |\mathbb{C}| > 0$, then there must be at least $|\mathbb{A}| - |\mathbb{C}|$ elements $\delta_i \in \mathbb{A} \cap \neg\mathbb{C}$, which satisfies the following constraints:

$$|\hat{c}_i - p_i^*| \;-\; \tau\,\mu(\hat{\boldsymbol{c}}, \boldsymbol{p}^*) \;>\; 0, \tag{25}$$

$$|c_i^* - p_i^*| \;-\; \tau\,\mu(\boldsymbol{c}^*, \boldsymbol{p}^*) \;\leq\; 0, \tag{26}$$

$$|\hat{c}_i - p_i^* - \delta_i| \;-\; \tau\,\mu(\hat{\boldsymbol{c}}, \hat{\boldsymbol{p}}) \;\leq\; 0 \quad \text{or} \quad |c_i^* - p_i^* - \delta_i| \;-\; \tau\,\mu(\boldsymbol{c}^*, \hat{\boldsymbol{p}}) \;>\; 0, \tag{27}$$

where the first two constraints follows from the definition of the set $\mathbb{A}$ in Equation 16 and the last constraint follows from the negation of the constraints in the set $\mathbb{C}$ in Equation 18. Now, we consider the two situations in Equation 27 separately:

① If $|\hat{c}_i - p_i^* - \delta_i| \;-\; \tau\,\mu(\hat{\boldsymbol{c}}, \hat{\boldsymbol{p}}) \;\leq\; 0$ in Equation 27, then we have:

$$|\hat{c}_i - p_i^* - \delta_i| \;-\; \tau\,\mu(\hat{\boldsymbol{c}}, \hat{\boldsymbol{p}}) \;\leq\; 0 \quad \text{and} \quad |\hat{y}_i - p_i^*| \;-\; \tau\,\mu(\hat{\boldsymbol{c}}, \boldsymbol{p}^*) \;>\; 0, \tag{28}$$

which in turn implies that:

$$|(|\hat{c}_i - p_i^*| - \tau \cdot \mu(\hat{\boldsymbol{c}}, \boldsymbol{p}^*)) - (|\hat{c}_i - (p_i^* + \delta_i)| - \tau \cdot \mu(\hat{\boldsymbol{c}}, \hat{\boldsymbol{p}}))| > ||\hat{c}_i - p_i^*| - \tau \cdot \mu(\hat{\boldsymbol{c}}, \boldsymbol{p}^*)|. \tag{29}$$

The inequality above can be rewritten (by swapping its two sides) as:

$$||\hat{c}_i - p_i^*| - \tau \cdot \mu(\hat{\boldsymbol{c}}, \hat{p}^*)| < |(|\hat{c}_i - p_i^*| - \tau \cdot \mu(\hat{\boldsymbol{c}}, \boldsymbol{p}^*)) - (|\hat{c}_i - (p_i^* + \delta_i)| - \tau \cdot \mu(\hat{\boldsymbol{c}}, \hat{\boldsymbol{p}}))|$$

$$\overset{(a)}{\leq} |(|\hat{c}_i - p_i^*| - \tau \cdot \mu(\hat{\boldsymbol{c}}, \boldsymbol{p}^*)) - ((|\hat{c}_i - p_i^*| - |\delta_i|) - \tau \cdot \mu(\hat{\boldsymbol{c}}, \hat{\boldsymbol{p}}))|$$

$$\overset{(b)}{=} ||\delta_i| - \tau \cdot (\mu(\hat{\boldsymbol{c}}, \boldsymbol{p}^*) - \mu(\hat{\boldsymbol{c}}, \hat{\boldsymbol{p}}))|$$

$$\overset{(c)}{=} \left| |\delta_i| - \tau \cdot \frac{1}{d} \sum_{j=1}^{d} (|\hat{c}_j - p_j^*| - |\hat{c}_j - \hat{p}_j|) \right|$$

$$\overset{(d)}{\leq} |\delta_i| + \tau \cdot \frac{1}{d} \sum_{j=1}^{d} \left| |\hat{c}_j - p_j^*| - |\hat{c}_j - \hat{p}_j| \right|$$

$$\overset{(e)}{\leq} |\delta_i| + \tau \cdot \frac{1}{d} \sum_{j=1}^{d} |\delta_j|$$

$$= |\delta_i| + \frac{\tau}{d} \cdot \|\boldsymbol{\delta}\|_1, \tag{30}$$

where $(a)$ follows from the fact that $|\hat{c}_i - (p_i^* + \delta_i)| \geq |\hat{c}_i - p_i^*| - |\delta_i|$, $(b)$ follows by reshuffling the terms in the inequality, $(c)$ follows from the definition of $\mu(\hat{c}, p^*)$ and $\mu(\hat{\boldsymbol{c}}, \hat{\boldsymbol{p}})$, $(d)$ follows from the triangle inequality, and $(e)$ follows from the definition of $\hat{p}_j = p_j^* + \delta_j$ and hence $|\hat{c}_j - \hat{p}_j| = |\hat{c}_j - p_j^* - \delta_j| < |\hat{c}_j - p_j^*| + |\delta_j|$ and $|\hat{c}_j - \hat{p}_j| = |\hat{c}_j - p_j^* - \delta_j| > |\hat{c}_j - p_j^*| - |\delta_j|$, which in turn implies that $\left| |\hat{c}_j - p_j^*| - |\hat{c}_j - \hat{p}_j| \right| \leq |\delta_j|$.

② Similarly, if $|c_i^* - p_i^* - \delta_i| \;-\; \tau\,\mu(\boldsymbol{c}^*, \hat{\boldsymbol{p}}) \;>\; 0$ in Equation 27, we get:

$$|c_i^* - p_i^* - \delta_i| \;-\; \tau\,\mu(\boldsymbol{c}^*, \hat{\boldsymbol{p}}) \;>\; 0 \quad \text{and} \quad |c_i^* - p_i^*| \;-\; \tau\,\mu(\boldsymbol{c}^*, \boldsymbol{p}^*) \;\leq\; 0 \tag{31}$$

which in turn implies that:

$$|(|c_i^* - p_i^*| - \tau \cdot \mu(\boldsymbol{c}^*, \boldsymbol{p}^*)) - (|c_i^* - (p_i^* + \delta_i)| - \tau \cdot \mu(\boldsymbol{c}^*, \hat{\boldsymbol{p}}))| > ||c_i^* - p_i^*| - \tau \cdot \mu(\boldsymbol{c}^*, \boldsymbol{p}^*)| \tag{32}$$

Following the same procedure in Equation 30, we conclude that:

$$||c_i^* - p_i^*| - \tau \cdot \mu(\boldsymbol{c}^*, \boldsymbol{p}^*)| < |(|c_i^* - p_i^*| - \tau \cdot \mu(\boldsymbol{c}^*, \boldsymbol{p}^*)) - (|c_i^* - (p_i^* + \delta_i)| - \tau \cdot \mu(\boldsymbol{c}^*, \hat{\boldsymbol{p}}))|$$

$$\leq |\delta_i| + \frac{\tau}{d} \cdot \|\boldsymbol{\delta}\|_1. \tag{33}$$

By combining situations ① and ② together, we conclude that if $|\mathbb{A}| - |\mathbb{C}| > 0$ then:

$$|\delta_i| + \frac{\tau}{d} \cdot \|\boldsymbol{\delta}\|_1 > ||c_i^* - p_i^*| - \tau \cdot \mu(\boldsymbol{c}^*, \boldsymbol{p}^*)| \tag{34}$$

$$\text{or} > ||\hat{c}_i - p_i^*| - \tau \cdot \mu(\hat{\boldsymbol{c}}, \boldsymbol{p}^*)|. \tag{35}$$

Which can be combined toghether in one condition as:

$$|\delta_i| \geq min \left\{ ||\hat{c}_i - p_i^*| - \tau \mu(\hat{\boldsymbol{c}}, \boldsymbol{p}^*)| - \frac{\tau ||\boldsymbol{\delta}||_1}{d}, ||c_i^* - p_i^*| - \tau \mu(\boldsymbol{c}^*, \boldsymbol{p}^*)| - \frac{\tau ||\boldsymbol{\delta}||_1}{d} \right\}.$$

• **Case 2:** If $|\mathbb{D}| - |\mathbb{B}| > 0$, then there must be at least $|\mathbb{D}| - |\mathbb{B}|$ elements $\delta_i \in \mathbb{D} \cap \neg\mathbb{B}$, which satisfies the following constraints:

$$|\hat{c}_i - p_i^* - \delta_i| - \tau \cdot \mu(\hat{\boldsymbol{c}}, \hat{\boldsymbol{p}}) \leq 0, \tag{36}$$

$$|c_i^* - p_i^* - \delta_i| - \tau \cdot \mu(\boldsymbol{c}^*, \hat{\boldsymbol{p}}) > 0 \tag{37}$$

$$|\hat{c}_i - p_i^*| - \tau \cdot \mu(\hat{\boldsymbol{c}}, \boldsymbol{p}^*) > 0 \quad \text{or} \quad |c_i^* - p_i^*| - \tau \cdot \mu(\boldsymbol{c}^*, \boldsymbol{p}^*) \leq 0, \tag{38}$$

where the first two constraints follows from the definition of the set $\mathbb{D}$ in Equation 19 and the last constraint follows from the negation of the constraints in the set $\mathbb{B}$ in Equation 17. Now, we consider the two situations in Equation 38 separately:

① If $|\hat{c}_i - p_i^*| - \tau \cdot \mu(\hat{\boldsymbol{c}}, \boldsymbol{p}^*) > 0$ in Equation 38, then we have:

$$|\hat{c}_i - p_i^*| - \tau \cdot \mu(\hat{\boldsymbol{c}}, \boldsymbol{p}^*) > 0 \quad \text{and} \quad |\hat{c}_i - p_i^* - \delta_i| - \tau \cdot \mu(\hat{\boldsymbol{c}}, \hat{\boldsymbol{p}}) \leq 0, \tag{39}$$

which in turn implies that:

$$|(|\hat{c}_i - p_i^*| - \tau \cdot \mu(\hat{\boldsymbol{c}}, \boldsymbol{p}^*)) - (|\hat{c}_i - (p_i^* + \delta_i)| - \tau \cdot \mu(\hat{\boldsymbol{c}}, \hat{\boldsymbol{p}}))| > ||\hat{c}_i - p_i^*| - \tau \cdot \mu(\hat{\boldsymbol{c}}, \boldsymbol{p}^*)|. \tag{40}$$

Following the same procedure in Equation 30, we conclude that:

$$||\hat{c}_i - p_i^*| - \tau \cdot \mu(\hat{\boldsymbol{c}}, \boldsymbol{p}^*)| < |(|\hat{c}_i - p_i^*| - \tau \cdot \mu(\hat{\boldsymbol{c}}, \boldsymbol{p}^*)) - (|\hat{c}_i - (p_i^* + \delta_i)| - \tau \cdot \mu(\hat{\boldsymbol{c}}, \hat{\boldsymbol{p}}))| \tag{41}$$

$$\leq |\delta_i| + \frac{\tau}{d} \cdot \|\boldsymbol{\delta}\|_1. \tag{42}$$

② Similarly, if $|c_i^* - p_i^* - \delta_i| - \tau \cdot \mu(\boldsymbol{c}^*, \hat{\boldsymbol{p}}) > 0$ in Equation 38, we get:

$$|c_i^* - p_i^*| - \tau \cdot \mu(\boldsymbol{c}^*, \boldsymbol{p}^*) \leq 0 \quad \text{and} \quad |c_i^* - p_i^* - \delta_i| - \tau \cdot \mu(\boldsymbol{c}^*, \hat{\boldsymbol{p}}) > 0, \tag{43}$$

which in turn implies that:

$$|(|c_i^* - p_i^*| - \tau \cdot \mu(\boldsymbol{c}^*, \boldsymbol{p}^*)) - (|c_i^* - (p_i^* + \delta_i)| - \tau \cdot \mu(\boldsymbol{c}^*, \hat{\boldsymbol{p}}))| > ||c_i^* - p_i^*| - \tau \cdot \mu(\boldsymbol{c}^*, \boldsymbol{p}^*)|. \tag{44}$$

Following the same procedure in Equation 30, we conclude that:

$$||c_i^* - p_i^*| - \tau \cdot \mu(\boldsymbol{c}^*, \boldsymbol{p}^*)| < |(|c_i^* - p_i^*| - \tau \cdot \mu(\boldsymbol{c}^*, \boldsymbol{p}^*)) - (|c_i^* - (p_i^* + \delta_i)| - \tau \cdot \mu(\boldsymbol{c}^*, \hat{\boldsymbol{p}}))|$$

$$\leq |\delta_i| + \frac{\tau}{d} \cdot \|\boldsymbol{\delta}\|_1. \tag{45}$$

By combining the two situations ① and ② together, we conclude that if $|\mathbb{D}| - |\mathbb{B}| > 0$ then:

$$|\delta_i| + \frac{\tau}{d} \cdot \|\boldsymbol{\delta}\|_1 > ||c_i^* - p_i^*| - \tau \cdot \mu(\boldsymbol{c}^*, \boldsymbol{p}^*)| \tag{46}$$

$$\text{or} > ||\hat{c}_i - p_i^*| - \tau \cdot \mu(\hat{\boldsymbol{c}}, \boldsymbol{p}^*)|. \tag{47}$$

Combining the two inequalities above, we conclude:

$$|\delta_i| \geq min\left\{ |\,|\hat{c}_i - p_i^*| \; - \; \tau\,\mu(\hat{\boldsymbol{c}}, \boldsymbol{p}^*)| - \frac{\tau||\boldsymbol{\delta}||_1}{d}, |c_i^* - p_i^*| \; - \; \tau\,\mu(\boldsymbol{c}^*, \boldsymbol{p}^*)| - \frac{\tau||\boldsymbol{\delta}||_1}{d} \right\}$$

which in turn implies that:

$$|\delta_i| \geq min\{|\,|\hat{c}_i - p_i^*| \; - \; \tau\,\mu(\hat{\boldsymbol{c}}, \boldsymbol{p}^*)| - \frac{\tau||\boldsymbol{\delta}||_1}{d}, |c_i^* - p_i^*| \; - \; \tau\,\mu(\boldsymbol{c}^*, \boldsymbol{p}^*)| - \frac{\tau||\boldsymbol{\delta}||_1}{d}\}.$$

Thus for both **Case 1** and **Case 2** we get the same conclusion that:

$$|\delta_i| \geq min\left\{ |\,|\hat{c}_i - p_i^*| \; - \; \tau\,\mu(\hat{\boldsymbol{c}}, \boldsymbol{p}^*)| - \frac{\tau||\boldsymbol{\delta}||_1}{d}, |c_i^* - p_i^*| \; - \; \tau\,\mu(\boldsymbol{c}^*, \boldsymbol{p}^*)| - \frac{\tau||\boldsymbol{\delta}||_1}{d} \right\}.$$

Thus there are in total at least $\Delta L_0(\boldsymbol{p}^*)$ elements $\delta_i$ that fulfills:

$$|\delta_i| \geq min\left\{ |\,|\hat{c}_i - p_i^*| \; - \; \tau\,\mu(\hat{\boldsymbol{c}}, \boldsymbol{p}^*)| - \frac{\tau||\boldsymbol{\delta}||_1}{d}, |c_i^* - p_i^*| \; - \; \tau\,\mu(\boldsymbol{c}^*, \boldsymbol{p}^*)| - \frac{\tau||\boldsymbol{\delta}||_1}{d} \right\}, \quad (48)$$

since $\Delta L_0(\boldsymbol{p}^*) - \Delta L_0(\hat{\boldsymbol{p}}) \geq \Delta L_0(\boldsymbol{p}^*)$. $\qquad\qquad\square$

### B.3 NECESSARY CONDITION FOR SUCCESSFUL ATTACK ON $KL$ AND $L_0$ DETECTORS ARE MUTUALLY EXCLUSIVE

**Proposition 4** (Necessary conditions of successful attacks on KL and $L_0$ detectors are mutually exclusive). *Let $\boldsymbol{p}^* \in \mathbb{R}^d$ be the input embedding (feature vector) of the clean image and $\hat{\boldsymbol{p}} \in \mathbb{R}^d$ be the input embedding of the adversarially attacked image, i.e., $\hat{\boldsymbol{p}} = \boldsymbol{p}^* + \boldsymbol{\delta}$ where $\boldsymbol{\delta}$ is the adversarial perturbation of the input embedding. Assume that the Assumptions A1-A4 are satisfied, then if a threshold $\tau$ exists such that:*

$$\frac{||\boldsymbol{v}||}{\epsilon} \sum_{i \in \mathbb{S}^{unchange}} (\delta_i^{max})^2 + \sum_{i \in \mathbb{S}^{change}} min_i\, v_i \; + \; \epsilon^{remain} \sqrt{\sum_{i \in \mathbb{S}^{remain}} v_i^2} < \Delta KL(\boldsymbol{p}^*), \qquad (49)$$

*then there exists no perturbation $\delta$ that can render $\hat{y}_{KL} = \hat{y}_{L_0}$, where:*

$$\Delta KL(\boldsymbol{p}^*) = KL(\hat{\boldsymbol{c}}||\boldsymbol{p}^*) - KL(\boldsymbol{c}^*||\boldsymbol{p}^*),$$
$$\Delta L_0(\boldsymbol{p}^*) = L_0(\hat{\boldsymbol{c}}||\boldsymbol{p}^*) - L_0(\boldsymbol{c}^*||\boldsymbol{p}^*),$$
$$min_i = \min\left\{ |\,|\hat{c}_i - p_i^*| - \tau\mu(\hat{\boldsymbol{c}}, \boldsymbol{p}^*)| - \frac{\tau||\boldsymbol{\delta}||_1}{d}, |c_i^* - p_i^*| - \tau\mu(\boldsymbol{c}^*, \boldsymbol{p}^*)| - \frac{\tau||\boldsymbol{\delta}||_1}{d}. \right\},$$
$$v_i = \frac{\hat{c}_i - c_i^*}{p_i*},$$
$$\delta_i^{max} = \epsilon \cdot \frac{v_i}{||\boldsymbol{v}||}$$
$$\mathbb{S}^{unchange} = \left\{ i \in \{1, \dots, m\} \,|\, |\delta_i^{max}| \geq min_i \right\},$$
$$\mathbb{S}^{change} = \underset{\mathbb{T} \subseteq \{1,\dots,m\} \setminus \mathbb{S}^{unchange}}{\arg\min} \sum_{i \in \mathbb{T}} |\delta_i^{max} - min_i|,$$
$$\mathbb{S}^{remain} = \{1, \dots, m\} \setminus (\mathbb{S}^{unchange} \cup \mathbb{S}^{change}),$$
$$\epsilon^{remain} = \sqrt{\epsilon^2 - \sum_{i \in \mathbb{S}^{change}} (min_i)^2 - \sum_{i \in \mathbb{S}^{unchange}} (\delta_i^{max})^2}.$$

*Proof.* Our proof focuses on establishing a contradiction by showing that no perturbation $\boldsymbol{\delta}$ can simultaneously satisfy both Proposition 2 and Proposition 3. Specifically, we will assume, for the sake of contradiction, that there exists a perturbation $\boldsymbol{\delta}'$ that satisfies the constraints required by Proposition 3, and we show that even under these constraints, **the maximum achievable value of $\boldsymbol{v}^T \boldsymbol{\delta}'$ cannot exceed** $\Delta KL(\boldsymbol{p}^*)$, contradicting the condition required by Proposition 2.

Finding such a $\boldsymbol{\delta}'$ is equivalent to solving the following constrained optimization problem:

$$\boldsymbol{\delta}' := \arg\max_{\boldsymbol{\delta}' \in \mathbb{R}^d} \quad \boldsymbol{v}^T \boldsymbol{\delta}'$$

$$\text{subject to} \quad \|\boldsymbol{\delta}'\|_2 = \epsilon, \tag{50}$$

$$|\delta_i'| \geq \min_i \quad \text{for at least } \Delta \mathrm{L}_0(\boldsymbol{p}^*) \text{ coordinates.}$$

where $v$ is the KL-gradient direction vector, i.e: $v_i := \frac{\hat{c}_i - c_i^*}{p_i^*}$. In other words, the optimization problem above aims to maximize the satisfaction of the condition imposed by Proposition 2 while satisfying the condition imposed by Proposition 3.

**Claim B.1.** *The solution of the optimization problem in Equation 50 can be obtained by following the following two-steps:*

• *Step 1: Solve the partially constrained maximization:*

$$\boldsymbol{\delta}^{\max} := \arg\max_{\|\boldsymbol{\delta}\|_2 = \epsilon} \boldsymbol{v}^T \boldsymbol{\delta}. \tag{51}$$

*This yields the perturbation that maximizes the dot product with $v$ under an $L_2$ norm constraint.*

• *Step 2: Project $\delta^{\max}$ onto the feasible set $\mathcal{C}$:*

$$\boldsymbol{\delta}' := \arg\min_{\boldsymbol{\delta} \in \mathcal{C}} \|\boldsymbol{\delta} - \boldsymbol{\delta}^{\max}\|_2^2, \tag{52}$$

*where $\mathcal{C}$ is the feasible set defined as:*

$$\mathcal{C} = \left\{ \boldsymbol{\delta} \in \mathbb{R}^d \,|\, \|\boldsymbol{\delta}'\|_2 = \epsilon, |\boldsymbol{\delta}_i'| \geq \min_i \quad \text{for at least } \Delta \mathrm{L}_0(\boldsymbol{p}^*) \text{ coordinates.} \right\}$$

We will provide a formal proof for Claim B.1 at the end of this section by comparing the KKT conditions for the optimization problem in Equation 50 with those from Equation 51 and Equation 52.

Based on Claim B.1, we proceed with the two steps above as follows. First, note that the $\boldsymbol{\delta}^{\max}$ is a maximizer for the inner product $\sum_i v_i \delta_i$ and hence the maximum is attained when the two vectors $v$ and $\boldsymbol{\delta}$ are aligned (i.e., the cosine of the angle between the two vectors is equal to 1). Second, note that any strictly interior point of $\|\boldsymbol{\delta}^{\mathbf{max}}\|_2 \leq \epsilon$ can be radially enlarged to increase $\boldsymbol{v}^\top \boldsymbol{\delta}^{\mathbf{max}}$, the maximum of Equation 50 must lie on the sphere $\|\boldsymbol{\delta}^{\mathbf{max}}\|_2 = \epsilon$. Hence, we conclude that:

$$\delta_i^{\max} = \frac{\epsilon}{\|\boldsymbol{v}\|_2} v_i. \tag{53}$$

Next, we find $\boldsymbol{\delta}'$ by projecting $\boldsymbol{\delta}^{\max}$ onto the constraint set $\mathcal{C}$ by solving:

$$\boldsymbol{\delta}' := \arg\min_{\boldsymbol{\delta} \in \mathcal{C}} \|\boldsymbol{\delta} - \boldsymbol{\delta}^{\max}\|_2^2.$$

To do so, we categorize the indices of $\boldsymbol{\delta}^{\max}$ into three groups and change them to $\boldsymbol{\delta}'$ accordingly by choosing the smallest possible $\Delta\delta_i$ on each dimension in order to minimize $\|\Delta\boldsymbol{\delta}' - \boldsymbol{\delta}^{\max}\|_2^2$. For sake of notation, we denote by $\min_i$ the requirement from Proposition 3 as: $\min_i = \min\left\{ ||\hat{c}_i - p_i^*| - \tau\mu(\hat{\boldsymbol{c}}, \boldsymbol{p}^*)| - \frac{\tau\|\boldsymbol{\delta}\|_1}{d}, ||c_i^* - p_i^*| - \tau\mu(\boldsymbol{c}^*, \boldsymbol{p}^*)| - \frac{\tau\|\boldsymbol{\delta}\|_1}{d} \right\}$.

We proceed as follows:

① For all indices $i$ where $|\delta_i^{\max}| \geq \min_i$, we add them to the set $\mathbb{S}^{\mathrm{unchange}}$, i.e., $\mathbb{S}^{\mathrm{unchange}} = \{i \in \{1, \ldots, m\} \,|\, |\delta_i^{\max}| \geq \min_i\}$. For this set, the corresponding $\delta_i'$ will be set to $\delta_i' = \delta_i^{\max}$ (i.e., their perturbation is unchanged).

② Recall that Proposition 3 requires a minimum of $\Delta L_0(\boldsymbol{p}^*)$ indices to have their perturbation higher than $\min_i$. Hence, we select $\Delta L_0(\boldsymbol{p}^*) - |\mathbb{S}^{\mathrm{unchange}}|$ elements whose $\delta_i^{\max} < \min_i$ and set the corresponding $\delta_i'$ to be $\delta_i' = \min_i$. Indeed, we select the indices $i$ whose $\delta_i^{\max}$ are as close as possible to $\min_i$, i.e.,

$$\mathbb{S}^{\mathrm{change}} = \arg\min_{\mathbb{T} \subseteq \{1, \ldots, m\} \setminus \mathbb{S}^{\mathrm{unchange}}} \sum_{i \in \mathbb{T}} |\delta_i^{\max} - \min_i|$$

③ For all remaining indices not in $\mathbb{S} = \mathbb{S}^{unchange} \cup \mathbb{S}^{change}$, we add them to the set $\mathbb{S}^{remain} = \{1, \ldots, m\} \setminus \mathbb{S}$, and we set the corresponding $\delta_i'$ according to the next Claim.

**Claim B.2.** *The solution of the optimization problem in Equation 50 $\delta'$ is:*

$$
\delta_i' = \begin{cases} min_i & \text{for } i \in \mathbb{S}^{change} \\ \delta_i^{\max} & \text{for } i \in \mathbb{S}^{unchange} \\ \epsilon^{remain} \cdot \dfrac{v_i}{\sqrt{\sum_{j \in \mathbb{S}^{remain}} v_j^2}} & \text{for } i \in \mathbb{S}^{remain} \end{cases}
\tag{54}
$$

*where:*

$$
\epsilon^{remain} = \sqrt{\epsilon^2 - \sum_{i \notin \mathbb{S}^{remain}} (\delta_i')^2} = \sqrt{\epsilon^2 - \sum_{i \in \mathbb{S}^{change}} (min_i)^2 - \sum_{i \in \mathbb{S}^{unchange}} (\delta_i^{\max})^2}.
$$

In this way, we keep as many elements as possible in $\delta'$ to make it close to $\delta^{max}$ while satisfying proposition 3. We will provide the proof of Claim B.2 at the end of this section.

To reach the contradiction, we need to show that the perturbation $\delta'$ violates Proposition 2, i.e., we would like to show that:

$$
\sum_i v_i \delta_i' \leq \Delta KL(p^*).
\tag{55}
$$

Define $\Delta \delta = \delta' - \delta^{\max}$ and substitute in the left hand side above as follows:

$$
\begin{aligned}
\sum_{i=1}^d v_i \delta_i' = v^T \delta' &= v^T (\delta^{\max} + \Delta \delta) \\
&\overset{(a)}{=} \frac{\|v\|}{\epsilon} (\delta^{\max})^T (\delta^{\max} + \Delta \delta) \\
&= \frac{\|v\|}{\epsilon} \left( \|\delta^{\max}\|_2^2 + (\delta^{\max})^T \Delta \delta \right) \\
&\overset{(b)}{=} \frac{\|v\|}{\epsilon} \left( \epsilon^2 + (\delta^{\max})^T \Delta \delta \right) \\
&\overset{(c)}{=} \frac{\|v\|}{\epsilon} \left( \epsilon^2 - \frac{1}{2} \|\Delta \delta\|_2^2 \right) \\
&= \|v\| \cdot \epsilon - \frac{\|v\|}{2\epsilon} \|\Delta \delta\|_2^2.
\end{aligned}
\tag{56}
$$

where $(a)$ follows from Equation 53 which states that $\delta^{\max} = \epsilon \cdot \frac{v}{\|v\|}$ and hence we can express $v$ as:

$$
v = \frac{\|v\|}{\epsilon} \delta^{\max} \quad \text{and thus } v^T = \frac{\|v\|}{\epsilon} (\delta^{\max})^T.
$$

The equalities $(b)$ and $(c)$ follows from the fact that both $\delta^{\max}$ and $\delta'$ satisfies the constraint $\|\delta^{\max}\|_2^2 = \|\delta'\|_2^2 = \epsilon^2$. Hence:

$$
\begin{aligned}
\|\delta'\|_2^2 - \|\delta^{\max}\|_2^2 = 0 &\Rightarrow \|\delta^{\max} + \Delta \delta\|_2^2 - \|\delta^{\max}\|_2^2 = 0 \\
&\Rightarrow \|\delta^{\max}\|_2^2 + 2(\delta^{\max})^T \Delta \delta + \|\Delta \delta\|_2^2 - \|\delta^{\max}\|_2^2 = 0 \\
&\Rightarrow 2(\delta^{\max})^T \Delta \delta + \|\Delta \delta\|_2^2 = 0 \\
&\Rightarrow (\delta^{\max})^T \Delta \delta = -\frac{1}{2} \|\Delta \delta\|_2^2
\end{aligned}
$$

Next, we expand the term $\|\Delta \delta\|_2^2$ in Equation 56 by substituting the values of $\delta'$ as:

$$
\|\Delta \delta\|_2^2 = \sum_{i \in \mathbb{S}^{change}} \left( min_i - \delta_i^{\max} \right)^2 + \sum_{i \in \mathbb{S}^{remain}} \left( b_i - \delta_i^{\max} \right)^2.
\tag{57}
$$

where $b_i = \epsilon^{remain} \cdot \dfrac{v_i}{\sqrt{\sum_{j \in \mathbb{S}^{remain}} v_j^2}}$

Expanding each quadratic and using $\sum_{i=1}^{d} (\delta_i^{\max})^2 = \epsilon^2$, one obtains:

$$\|\Delta\boldsymbol{\delta}\|_2^2 = \sum_{i\in\mathbb{S}^{\text{change}}} \left(\min_i^2 - 2\min_i \delta_i^{\max} + (\delta_i^{\max})^2\right) + \sum_{i\in\mathbb{S}^{\text{remain}}} \left(b_i^2 - 2b_i \delta_i^{\max} + (\delta_i^{\max})^2\right)$$

$$= \sum_{i\in\mathbb{S}^{\text{change}}} \min_i^2 + \sum_{i\in\mathbb{S}^{\text{remain}}} b_i^2 - 2\sum_{i=1}^{d} a_i^* \delta_i^{\max} + \sum_{i\in\mathbb{S}^{\text{change}}} (\delta_i^{\max})^2 + \sum_{i\in\mathbb{S}^{\text{remain}}} (\delta_i^{\max})^2$$

$$= \underbrace{\sum_{i\in\mathbb{S}^{\text{change}}} \min_i^2 + (\epsilon^{\text{remain}})^2}_{=\epsilon^2 - \sum_{i\in\mathbb{S}^{\text{unchange}}}(\delta_i^{\max})^2} - 2\sum_{i=1}^{d} a_i^* \delta_i^{\max} + \underbrace{\sum_{i\in\mathbb{S}^{\text{change}}} (\delta_i^{\max})^2 + \sum_{i\in\mathbb{S}^{\text{remain}}} (\delta_i^{\max})^2}_{=\epsilon^2 - \sum_{i\in\mathbb{S}^{\text{unchange}}}(\delta_i^{\max})^2}$$

$$= 2\left(\epsilon^2 - \sum_{i\in\mathbb{S}^{\text{unchange}}} (\delta_i^{\max})^2\right) - 2\sum_{i=1}^{d} a_i^* \delta_i^{\max}, \tag{58}$$

where:

$$a_i^* = \begin{cases} \min_i, & i \in \mathbb{S}^{\text{change}}, \\ 0, & i \in \mathbb{S}^{\text{unchange}}, \\ b_i, & i \in \mathbb{S}^{\text{remain}}. \end{cases}$$

We expand the term $\sum_{i=1}^{d} a_i^* \delta_i^{\max}$ in Equation 58 as follows:

$$\sum_{i=1}^{d} a_i^* \delta_i^{\max} = \sum_{i\in\mathbb{S}^{\text{change}}} \min_i \delta_i^{\max} + \sum_{i\in\mathbb{S}^{\text{remain}}} b_i \delta_i^{\max}$$

$$= \sum_{i\in\mathbb{S}^{\text{change}}} \min_i \frac{\epsilon v_i}{\|\boldsymbol{v}\|_2} + \sum_{i\in\mathbb{S}^{\text{remain}}} \left(\epsilon^{\text{remain}} \frac{\epsilon |v_i|^2}{\|\boldsymbol{v}\|_2 \sqrt{\sum_{j\in\mathbb{S}^{\text{remain}}} v_j^2}}\right)$$

$$= \frac{\epsilon}{\|\boldsymbol{v}\|_2}\left(\sum_{i\in\mathbb{S}^{\text{change}}} \min_i v_i + \epsilon^{\text{remain}} \sqrt{\sum_{i\in\mathbb{S}^{\text{remain}}} v_i^2}\right). \tag{59}$$

Substituting back in Equation 58 we conclude:

$$\|\Delta\boldsymbol{\delta}\|_2^2 = 2\epsilon^2 - 2\sum_{i\in\mathbb{S}^{\text{unchange}}} (\delta_i^{\max})^2 - \frac{2\epsilon}{\|\boldsymbol{v}\|_2}\left[\sum_{i\in S}\min_i v_i + \epsilon^{\text{remain}} \sqrt{\sum_{i\in\mathbb{S}^{\text{remain}}} v_i^2}\right]$$

$$\overset{(d)}{>} 2\epsilon^2 - \frac{2\epsilon}{\|\boldsymbol{v}\|}\Delta KL(\boldsymbol{p}^*), \tag{60}$$

where $(d)$ follows from the assumption on $\tau$ in Equation 49 which requires that:

$$\frac{\|\boldsymbol{v}\|}{\epsilon} \sum_{i\in\mathbb{S}^{\text{unchange}}} (\delta_i^{\max})^2 + \sum_{i\in\mathbb{S}^{\text{change}}} \min_i v_i + \epsilon^{\text{remain}} \sqrt{\sum_{i\in\mathbb{S}^{\text{remain}}} v_i^2} < \Delta KL(\boldsymbol{p}^*),$$

Finally, by combining the equation above with Equation 56, we arrive at:

$$\sum_{i=1}^{d} v_i \delta_i' = |v\| \cdot \epsilon - \frac{\|\boldsymbol{v}\|}{2\epsilon}\|\Delta\boldsymbol{\delta}\|_2^2 < \Delta KL(\boldsymbol{p}^*),$$

which contradicts Proposition 2.

To finalize our proof, we need to show that Claim B.1 and Claim B.2 holds.

• **Proof of Claim B.1:** We proceed by showing the equivalence between the KKT conditions for the two optimization problems as follows.

**KKT Conditions for the optimization problem in Equation 50.** By introducing the Lagrangian multiplier $\lambda \geq 0$ for $\|\boldsymbol{\delta}\|_2 = \epsilon$ and $\mu_i \geq 0$ for the $|\delta_i| \geq min_i$ bounds (active set $\mathbb{S}$ with $|\mathbb{S}| = \Delta L_0(p^*)$), we can write the Lagrangian as:

$$\mathcal{L}_{P1}(\boldsymbol{\delta}, \lambda, \mu) = -v^\top \boldsymbol{\delta} + \lambda\big(\|\boldsymbol{\delta}\|_2^2 - \epsilon^2\big) + \sum_{i \in S} \mu_i\big(\texttt{min}_i - |\delta_i|\big).$$

The corresponding KKT conditions are:

$$\frac{\partial \mathcal{L}}{\partial \delta_j} = -v_j + 2\lambda\, \delta_j - \mathbf{1}_{[j \in S]}\, \mu_j\, \text{sgn}(\delta_j) = 0, \tag{KKT-1}$$

$$\lambda\big(\|\boldsymbol{\delta}\|_2^2 - \epsilon^2\big) = 0, \tag{KKT-2}$$

$$\mu_i\big(|\delta_i| - \texttt{min}_i\big) = 0 \quad \text{for} \quad i \in S. \tag{KKT-3}$$

for each coordinate $j = 1, \ldots, d$, where $sgn(\cdot)$ is sign function. Note that $\|\boldsymbol{\delta}\|_2 = \epsilon$ and hence $(\|\boldsymbol{\delta}\|_2^2 - \epsilon^2) = 0$, implying that $\lambda \in \mathbb{R}$. For the free coordinates $(j \notin \mathbb{S})$, we obtain:

$$\delta_j^\star = \frac{v_j}{2\lambda}.$$

While for $i \in \mathbb{S}$ there are two cases: inactive bound $(\mu_i = 0)$ which gives the same expression as above and active bound $(|\delta_i^\star| = \texttt{min}_i, \mu_i > 0)$ which yields:

$$\delta_i^\star = \frac{v_i}{2\lambda} + \frac{\mu_i}{2\lambda}\, \text{sgn}(\delta_i^\star).$$

**KKT Conditions for the for the optimization problems in Equation 51 and Equation 52.**
For the optimization problem in Equation 51, our analysis above (Equation 53) shows that:

$$\boldsymbol{\delta}^{\max} = \frac{\epsilon}{\|\boldsymbol{v}\|_2} v.$$

Hence, we focus on obtaining the KKT conditions for the optimization problem in Equation 52. We start by constructing its Lagrangian (with the multipliers $\tilde{\lambda} \in \mathbb{R}$ and $\tilde{\mu}_i \geq 0$) as:

$$\mathcal{L}_{P2}(\boldsymbol{\delta}, \tilde{\lambda}, \mu) = \tfrac{1}{2}\|\boldsymbol{\delta} - \boldsymbol{\delta}^{\max}\|_2^2 + \tilde{\lambda}\big(\|\boldsymbol{\delta}\|_2^2 - \epsilon^2\big) + \sum_{i \in S} \tilde{\mu}_i\big(\texttt{min}_i - |\delta_i|\big).$$

The resulting KKT conditions are then:

$$\frac{\partial \mathcal{L}}{\partial \delta_j} = (\boldsymbol{\delta} - \boldsymbol{\delta}^{\max})_j + 2\tilde{\lambda}\, \delta_j - \mathbf{1}_{[j \in S]}\, \tilde{\mu}_j\, \text{sgn}(\delta_j) = 0, \tag{KKT$'$-1}$$

$$\tilde{\lambda}\big(\|\boldsymbol{\delta}\|_2^2 - \epsilon^2\big) = 0, \tag{KKT$'$-2}$$

$$\tilde{\mu}_i\big(|\delta_i| - \texttt{min}_i\big) = 0 \quad \text{for} \quad i \in \mathbb{S}. \tag{KKT$'$-3}$$

for each coordinate $j = 1, \ldots, d$. Let $\frac{(1+2\tilde{\lambda})\|\boldsymbol{v}\|}{\epsilon} = 2\lambda$ and $\tilde{\mu} = \frac{\epsilon}{\|\boldsymbol{v}\|}\mu$ (if $\mu \geq 0$ then $\tilde{\mu} \geq 0$), then the KKT$'$-1 can be written as:

$$\begin{aligned}
\frac{\partial \mathcal{L}}{\partial \delta_j} &= \underbrace{(\boldsymbol{\delta} - \boldsymbol{\delta}^{\max})_j + 2\tilde{\lambda}\, \delta_j - \mathbf{1}_{[j \in S]}\, \tilde{\mu}_j\, \text{sgn}(\delta_j)}_{KKT'-1} \\
&= (1 + 2\tilde{\lambda})\delta_j - \delta_j^{max} - \mathbf{1}_{[j \in S]}\, \tilde{\mu}_j\, \text{sgn}(\delta_j) \\
&= (1 + 2\tilde{\lambda})\delta_j - \frac{\epsilon}{\|\boldsymbol{v}\|_2} v_j - \mathbf{1}_{[j \in S]}\, \mu_j\, \text{sgn}(\delta_j) \\
&= \frac{\epsilon}{\|\boldsymbol{v}\|}\left((1 + 2\tilde{\lambda})\frac{\|\boldsymbol{v}\|}{\epsilon}\delta_j - v_j\right) - \mathbf{1}_{[j \in S]}\, \tilde{\mu}_j\, \text{sgn}(\delta_j) \\
&\overset{(a)}{=} \frac{\epsilon}{\|\boldsymbol{v}\|}(2\lambda\delta_j - v_j) - \mathbf{1}_{[j \in S]}\, \tilde{\mu}_j\, \text{sgn}(\delta_j) \\
&\overset{(b)}{=} \frac{\epsilon}{\|\boldsymbol{v}\|}\underbrace{(2\lambda\delta_j - v_j - \mathbf{1}_{[j \in S]}\, \mu_j\, \text{sgn}(\delta_j))}_{KKT-1} = 0. \tag{61}
\end{aligned}$$

where $(a)$ follows from the equality $\frac{(1+2\tilde{\lambda})\|\boldsymbol{v}\|}{\epsilon} = 2\lambda$ and $(b)$ follows from the equality $\tilde{\mu} = \frac{\epsilon}{\|\boldsymbol{v}\|}\mu$. Similarly, KKT-2$'$ is:

$$\tilde{\lambda}(\|\boldsymbol{\delta}\|_2^2 - \epsilon^2) = \underbrace{(\frac{\epsilon\lambda}{\|\boldsymbol{v}\|} - \frac{1}{2})}_{\in \mathbb{R}}(\|\boldsymbol{\delta}\|_2^2 - \epsilon^2) = 0. \tag{62}$$

And, KKT-3$'$ is:

$$\tilde{\mu}_i(|\delta_i| - m_i) = \frac{\epsilon}{\|\boldsymbol{v}\|}\underbrace{\mu(|\delta_i| - \mathtt{min}_i)}_{KKT-3} = 0 \quad \text{for} \quad i \in S. \tag{63}$$

From which we conclude that the optimal $\boldsymbol{\delta}'$ derived by solving KKT$'$-1-3 are *identical* to that derived from KKT-1-3, which in turn implies that the optimization problems in Equation 51 and Equation 52 are equivalent.

$\square$

• **Proof of Claim C.2:**

To obtain the value of $\boldsymbol{\delta}'_{\text{remain}}$, we aim to minimize the $\ell_2$-distance between $\boldsymbol{\delta}'_{\text{remain}}$ and the corresponding part of $\boldsymbol{\delta}^{\max}$, that is:

$$\min_{\boldsymbol{\delta}'_{\text{remain}}} \sum_{i \in \mathbb{S}^{\text{remain}}} (\boldsymbol{\delta}'_i - \boldsymbol{\delta}_i^{\max})^2,$$

$$\text{subject to:} \quad \sum_{i \in \mathbb{S}^{\text{remain}}} (\delta'_i)^2 = \epsilon^2 - \sum_{i \in \mathbb{S}^{\text{change}} \cup \mathbb{S}^{\text{unchange}}} (\delta'_i)^2$$

We solve this optimization problem using Lagrangian multiplier. We define the Lagrangian as:

$$\mathcal{L}(\boldsymbol{\delta}'_{\text{remain}}, \lambda) = (\boldsymbol{\delta}'_{\text{remain}} - \boldsymbol{\delta}_{\text{remain}}^{\max})^T(\boldsymbol{\delta}'_{\text{remain}} - \boldsymbol{\delta}_{\text{remain}}^{\max}) + \lambda\left((\boldsymbol{\delta}'_{\text{remain}})^T\boldsymbol{\delta}'_{\text{remain}} - (\epsilon^{\text{remain}})^2\right).$$

We calculate the gradient with respect to $\boldsymbol{\delta}'_{\text{remain}}$ and set it to zero as:

$$\nabla_{\boldsymbol{\delta}'_{\text{remain}}}\mathcal{L} = 2(\boldsymbol{\delta}'_{\text{remain}} - \boldsymbol{\delta}_{\text{remain}}^{\max}) + 2\lambda\boldsymbol{\delta}'_{\text{remain}} = 0.$$

Divide by 2 and rearrange:

$$(1 + \lambda)\boldsymbol{\delta}'_{\text{remain}} = \boldsymbol{\delta}_{\text{remain}}^{\max} \quad \Rightarrow \quad \boldsymbol{\delta}'_{\text{remain}} = \frac{1}{1 + \lambda}\boldsymbol{\delta}_{\text{remain}}^{\max}.$$

Due to the constraint $\|\boldsymbol{\delta}'_{\text{remain}}\|_2 = \epsilon^{\text{remain}}$, with $\epsilon^{\text{remain}} = \sqrt{\epsilon^2 - \sum_{i \notin \mathbb{S}^{\text{remain}}}(\delta'_i)^2}$, we get:

$$\left\|\frac{1}{1 + \lambda}\boldsymbol{\delta}_{\text{remain}}^{\max}\right\|_2 = \epsilon^{\text{remain}} \quad \Rightarrow \quad \frac{1}{|1 + \lambda|}\|\boldsymbol{\delta}_{\text{remain}}^{\max}\|_2 = \epsilon^{\text{remain}}.$$

Since $1 + \lambda > 0$:

$$1 + \lambda = \frac{\|\boldsymbol{\delta}_{\text{remain}}^{\max}\|_2}{\epsilon^{\text{remain}}} \quad \Rightarrow \quad \boldsymbol{\delta}'_{\text{remain}} = \frac{\epsilon^{\text{remain}}}{\|\boldsymbol{\delta}_{\text{remain}}^{\max}\|_2}\boldsymbol{\delta}_{\text{remain}}^{\max}.$$

Now using the expression $\boldsymbol{\delta}_i^{\max} = \frac{\epsilon}{\|\boldsymbol{v}\|}v_i$, we compute:

$$\|\boldsymbol{\delta}_{\text{remain}}^{\max}\|_2 = \frac{\epsilon}{\|\boldsymbol{v}\|}\sqrt{\sum_{i \in \mathbb{S}^{\text{remain}}}|v_i|^2}$$

Thus,

$$\delta'_i = \frac{\epsilon^{\text{remain}}}{\frac{\epsilon}{\|\boldsymbol{v}\|}\sqrt{\sum_{i \in \mathbb{S}^{\text{remain}}}|v_i|^2}} \cdot \frac{\epsilon}{\|\boldsymbol{v}\|}v_i = \frac{\epsilon^{\text{remain}}v_i}{\sqrt{\sum_{j \in \mathbb{S}^{\text{remain}}}v_j^2}} \quad \text{if} \quad i \in \mathbb{S}^{\text{remain}}.$$

$\square$

### B.4 PROOF OF THEOREM 1

*Proof of Theorem 1.* Since Theorem 1 asks for a minimum of one dimension for which the gap between the two prototype vectors $|\hat{c}_i - \hat{c}_i^*|$ is large enough, i.e., $|\hat{c}_i - \hat{c}_i^*| > \Gamma(\epsilon)$ for some threshold $\Gamma(\epsilon)$, we consider the extreme condition when such condition is satisfied for only one dimension. In such scenario, $\Delta L_0(p^*) = 1$ $|\mathbb{S}| = 1$, and $|\mathbb{S}^{\text{change}}| = 1, |\mathbb{S}^{\text{unchange}}| = 0$, $|\mathbb{S}^{\text{remain}}| = d - 1$ and $\epsilon^{\text{remain}} = \sqrt{\epsilon^2 - \min_i^2}$. The remainder of this proof follows two steps. First, we rewrite the condition on $\tau$ Equation 49 into an explicit form. Note that the parameter $\tau$ controls the separation between the classes in the $L_0$ sense that is needed to distinguish between the classes. Second, we derive a lower bound on the threshold $\Gamma(\epsilon)$ that guarantees the existence of the parameter $\tau$.

Since $|\mathbb{S}^{\text{unchange}}| = 0$, we can rewrite the condition on $\tau$ Equation 49 into:

$$\sum_{j \in \mathbb{S}^{\text{change}}} \min_j v_j + \epsilon^{\text{remain}} \sqrt{\sum_{i \in \mathbb{S}^{\text{remain}}} v_i^2} < \Delta KL(\boldsymbol{p}^*)$$

$$\overset{(a)}{\Rightarrow} \min_j v_j + \sqrt{\epsilon^2 - \min_j^2} \sqrt{\sum_{i \neq j} v_i^2} < \Delta KL(\boldsymbol{p}^*)$$

$$\overset{(b)}{\Rightarrow} \sqrt{\epsilon^2 - \min_j^2} \sqrt{\sum_{i \neq j} v_i^2} < \Delta KL(\boldsymbol{p}^*) - \min_j v_j$$

$$\overset{(c)}{\Rightarrow} (\epsilon^2 - \min_j^2)(\sum_{i \neq j} v_i^2) < (\Delta KL(\boldsymbol{p}^*) - \min_j v_j)^2$$

$$\Rightarrow (\epsilon^2 - \min_j^2)(\sum_{i \neq j} v_i^2) < (\Delta KL(\boldsymbol{p}^*))^2 - 2\Delta KL(\boldsymbol{p}^*)\min_j v_j + (\min_j v_j)^2$$

$$\overset{(d)}{\Rightarrow} (\min_j)^2(v_j^2 + \sum_{i \neq j} v_i^2) - 2\Delta KL(\boldsymbol{p}^*)\min_j v_j + (\Delta KL(\boldsymbol{p}^*)^2 - \epsilon^2 \sum_{i \neq j} v_i^2) > 0,$$

$$(64)$$

where $(a)$ follows from $|\mathbb{S}^{\text{unchange}}| = 0$ and $\epsilon^{\text{remain}} = \sqrt{\epsilon^2 - \min_j^2}$, $(b)$ follows from rearranging the terms, $(c)$ follows from squaring the two sides of the inequality, and $(d)$ follows from rearranging the terms.

We solve the quadratic equation below for the dummy variable $\mathbf{a}$:

$$(\mathbf{a})^2(v_j^2 + \sum_{i \neq j} v_i^2) - 2\Delta KL(\boldsymbol{p}^*)\mathbf{a}v_j + (\Delta KL(\boldsymbol{p}^*)^2 - \epsilon^2 \sum_{i \neq j} v_i^2) = 0 \tag{65}$$

which yields:

$$\mathbf{a} = \frac{\Delta KL(\boldsymbol{p}^*)v_j \pm \sqrt{\Delta KL(\boldsymbol{p}^*)^2(v_j)^2 - (v_j^2 + \sum_{i \neq j} v_i^2)(\Delta KL(\boldsymbol{p}^*)^2 - \epsilon^2 \sum_{i \neq j} v_i^2)}}{v_j^2 + \sum_{i \neq j} v_i^2}$$

$$(66)$$

Next, we expend the items inside the square root as:

$$\mathbf{a} = \frac{\Delta KL(\boldsymbol{p}^*)v_j}{v_j^2 + \sum_{i \neq j} v_i^2}$$

$$\pm \frac{\sqrt{\Delta KL(\boldsymbol{p}^*)^2(v_j)^2 - v_j^2 \Delta KL(\boldsymbol{p}^*)^2 - \sum_{i \neq j} v_i^2 \Delta KL(\boldsymbol{p}^*)^2 + v_j^2 \epsilon^2 \sum_{i \neq j} v_i^2 + \epsilon^2 (\sum_{i \neq j} v_i^2)^2}}{v_j^2 + \sum_{i \neq j} v_i^2}$$

$$= \frac{\Delta KL(\boldsymbol{p}^*)v_j \pm \sqrt{-\sum_{i \neq j} v_i^2 \Delta KL(\boldsymbol{p}^*)^2 + v_j^2 \epsilon^2 \sum_{i \neq j} v_i^2 + \epsilon^2 (\sum_{i \neq j} v_i^2)^2}}{\sum_{i=1}^d v_i^2},$$

where the last equality follows from the fact that $\Delta KL(\boldsymbol{p}^*)^2(v_j)^2 - v_j^2 \Delta KL(\boldsymbol{p}^*)^2 = 0$. We extract $\sqrt{\sum_{j\neq i} v_i^2}$ from square root as:

$$
\begin{aligned}
\mathrm{a} &= \frac{\Delta KL(\boldsymbol{p}^*)v_j \pm \sqrt{\sum_{i\neq j} v_i^2}\sqrt{-\Delta KL(\boldsymbol{p}^*)^2 + v_j^2\epsilon^2 + \epsilon^2\sum_{i\neq j} v_i^2}}{\sum_{i=1}^d v_i^2} \\
&= \frac{\Delta KL(\boldsymbol{p}^*)v_j \pm \sqrt{\sum_{i\neq j} v_i^2}\sqrt{-\Delta KL(\boldsymbol{p}^*)^2 + \epsilon^2(v_j^2 + \sum_{i\neq j} v_i^2)}}{\sum_{i=1}^d v_i^2} \\
&= \frac{\Delta KL(\boldsymbol{p}^*)v_j \pm \sqrt{\sum_{j\neq i} v_i^2}\sqrt{-\Delta KL(\boldsymbol{p}^*)^2 + \epsilon^2(\sum_{i=1}^d v_i^2)}}{\sum_{i=1}^d v_i^2} \\
&= \frac{\Delta KL(\boldsymbol{p}^*)v_j \pm \sqrt{\sum_{j\neq i} v_i^2}\sqrt{-\Delta KL(\boldsymbol{p}^*)^2 + \epsilon^2\|\boldsymbol{v}\|_2^2}}{\|\boldsymbol{v}\|_2^2}.
\end{aligned}
$$

As shown in the proof of Proposition 4, $\Delta KL(\boldsymbol{p}^*) < \epsilon\|\boldsymbol{v}\|$, we have $-\Delta KL(\boldsymbol{p}^*)^2 + \epsilon^2\|\boldsymbol{v}\|_2^2 > 0$, thus

$$
\mathrm{a}_1 = \frac{\Delta KL(\boldsymbol{p}^*)v_j - \sqrt{\sum_{j\neq i} v_i^2}\sqrt{-\Delta KL(\boldsymbol{p}^*)^2 + \epsilon^2\|\boldsymbol{v}\|_2^2}}{\|\boldsymbol{v}\|_2^2}
$$

or

$$
\mathrm{a}_2 = \frac{\Delta KL(\boldsymbol{p}^*)v_j + \sqrt{\sum_{j\neq i} v_i^2}\sqrt{-\Delta KL(\boldsymbol{p}^*)^2 + \epsilon^2\|\boldsymbol{v}\|_2^2}}{\|\boldsymbol{v}\|_2^2}.
$$

However,

$$
\begin{aligned}
\mathrm{a}_1 &= \frac{\Delta KL(\boldsymbol{p}^*)v_j - \sqrt{\sum_{j\neq i} v_i^2}\sqrt{-\Delta KL(\boldsymbol{p}^*)^2 + \epsilon^2\|\boldsymbol{v}\|_2^2}}{\|\boldsymbol{v}\|_2^2} < \frac{\Delta KL(\boldsymbol{p}^*)v_j}{\|\boldsymbol{v}\|_2^2} \\
&< \frac{\epsilon\|\boldsymbol{v}\|_2 v_j}{\|\boldsymbol{v}\|_2^2} = \frac{\epsilon v_j}{\|\boldsymbol{v}\|_2} = \delta_j^{\max}
\end{aligned}
$$

However, it follows from the definition of $\mathbb{S}^{\text{unchange}}$ in Proposition 4 that the $\min_j$ must fulfill that $\min_j > |\delta_j^{\max}|$. Hence, we conclude that $a_1$ is not a valid solution. Since $a_2$ is the only viable solution for the Equation 65, we conclude that the solution of the corresponding inequality (Equation 64) must satisfy:

$$
\min_j > \frac{\Delta KL(\boldsymbol{p}^*)v_j + \sqrt{\sum_{j\neq i} v_i^2}\sqrt{-\Delta KL(\boldsymbol{p}^*)^2 + \epsilon^2\|\boldsymbol{v}\|_2^2}}{\|\boldsymbol{v}\|_2^2}. \tag{67}
$$

Nevertheless, it follows from Proposition 4 that for each $j \in \mathbb{S}^{change} \subseteq \mathbb{S}$ this dimension fulfills:

$$
\begin{aligned}
|\delta_j^{\max}| &= \frac{\epsilon|v_j|}{\|\boldsymbol{v}\|} \\
&\leq \min\left\{\left||\hat{c}_j - p_j^*| - \tau\mu(\hat{\boldsymbol{c}}, \boldsymbol{p}^*)\right| - \frac{\tau\|\boldsymbol{\delta}\|_1}{d}, \left||c_j^* - p_j^*| - \tau\mu(\boldsymbol{c}^*, \boldsymbol{p}^*)\right| - \frac{\tau\|\boldsymbol{\delta}\|_1}{d}\right\} \\
&= \min_j \tag{68}
\end{aligned}
$$

Combining Equation 68 and Equation 67 then we can have

$$
\begin{aligned}
\min_j &= \min\left\{\left||\hat{c}_j - p_j^*| - \tau\mu(\hat{\boldsymbol{c}}, \boldsymbol{p}^*)\right| - \frac{\tau\|\boldsymbol{\delta}\|_1}{d}, \left||c_j^* - p_j^*| - \tau\mu(\boldsymbol{c}^*, \boldsymbol{p}^*)\right| - \frac{\tau\|\boldsymbol{\delta}\|_1}{d}\right\} \\
&> \max\left\{\frac{\epsilon|v_j|}{\|\boldsymbol{v}\|_2}, \frac{\Delta KL(\boldsymbol{p}^*)v_j + \|\boldsymbol{v}\|_2\sqrt{-\Delta KL(\boldsymbol{p}^*)^2 + \epsilon^2\|\boldsymbol{v}\|_2^2}}{\|\boldsymbol{v}\|_2^2}\right\}. \tag{69}
\end{aligned}
$$

Solving the inequality above for $\tau$, we conclude that we can rewrite the condition on $\tau$ from Equation 49 as:

$$\tau \leq \max\left\{ \frac{|\hat{c}_j - p_j^*| - \max\left\{ \frac{\epsilon|v_j|}{\|\boldsymbol{v}\|_2}, \frac{\Delta KL(\boldsymbol{p}^*)v_j + \|\boldsymbol{v}\|_2\sqrt{-\Delta KL(\boldsymbol{p}^*)^2 + \epsilon^2\|\boldsymbol{v}\|_2^2}}{\|\boldsymbol{v}\|_2^2} \right\}}{\mu(\hat{\boldsymbol{c}}, \boldsymbol{p}^*) + \frac{\|\boldsymbol{\delta}\|_1}{d}}, \right.$$

$$\left. \frac{|c_j^* - p_j^*| - \max\left\{ \frac{\epsilon|v_j|}{\|\boldsymbol{v}\|_2}, \frac{\Delta KL(\boldsymbol{p}^*)v_j + \|\boldsymbol{v}\|_2\sqrt{-\Delta KL(\boldsymbol{p}^*)^2 + \epsilon^2\|\boldsymbol{v}\|_2^2}}{\|\boldsymbol{v}\|_2^2} \right\}}{\mu(\boldsymbol{c}^*, \boldsymbol{p}^*) + \frac{\|\boldsymbol{\delta}\|_1}{d}} \right\}. \tag{70}$$

Note that while Equation 49 was an implicit constraint on $\tau$, the constraint above is an explicit constraint on $\tau$. Next, we derive the condition on the gap $|\hat{c}_j - c_i^j|$ that ensures that Equation 70 has a non-empty set of solutions.

It follows from Proposition 3 and the fact that we are considering the case where only one dimension satisfies the class gap $|\hat{c}_i - c_i^*|$ that such dimension $j$ must belong to the set $\mathbb{A}$ where: $\mathbb{A} = \{i : |\hat{c}_i - p_i^*| - \tau \cdot \mu(\hat{\boldsymbol{c}}, \boldsymbol{p}^*) > 0, |c_i^* - p_i^*| - \tau \cdot \mu(\boldsymbol{c}^*, \boldsymbol{p}^*) \leq 0\}$, and hence $\tau$ must satisfy the two constraints imposed by the set $\mathbb{A}$, i.e.,

$$\frac{|c_j^* - p_j^*|}{\mu(\boldsymbol{c}^*, \boldsymbol{p}^*)} < \tau < \frac{|\hat{c}_j - p_j^*|}{\mu(\hat{\boldsymbol{c}}, \boldsymbol{p}^*)} \tag{71}$$

Note that:

$$\frac{|\hat{c}_i - p_i^*| - \max\left\{ \frac{\epsilon|v_j|}{\|\boldsymbol{v}\|_2}, \frac{\Delta KL(\boldsymbol{p}^*)v_j + \|\boldsymbol{v}\|_2\sqrt{-\Delta KL(\boldsymbol{p}^*)^2 + \epsilon^2\|\boldsymbol{v}\|_2^2}}{\|\boldsymbol{v}\|_2^2} \right\}}{\mu(\hat{\boldsymbol{c}}, \boldsymbol{p}^*) + \frac{\|\boldsymbol{\delta}\|_1}{d}} < \frac{|\hat{c}_j - p_j^*|}{\mu(\hat{\boldsymbol{c}}, \boldsymbol{p}^*)}$$

and:

$$\frac{|c_i^* - p_i^*| - \max\left\{ \frac{\epsilon|v_j|}{\|\boldsymbol{v}\|_2}, \frac{\Delta KL(\boldsymbol{p}^*)v_j + \|\boldsymbol{v}\|_2\sqrt{-\Delta KL(\boldsymbol{p}^*)^2 + \epsilon^2\|\boldsymbol{v}\|_2^2}}{\|\boldsymbol{v}\|_2^2}, \sqrt{\frac{5}{4}\epsilon^2 - \frac{\epsilon\Delta KL(\boldsymbol{p}^*)}{\|\boldsymbol{v}\|}} \right\}}{\mu(\boldsymbol{c}^*, \boldsymbol{p}^*) + \frac{\|\boldsymbol{\delta}\|_1}{d}} < \frac{|c_j^* - p_j^*|}{\mu(\boldsymbol{c}^*, \boldsymbol{p}^*)}$$

since both the have smaller nominator and a larger denominator on the left-hand-side.

Thus we can write Equation 71 and Equation 70 together as:

$$\frac{|c_j^* - p_j^*|}{\mu(\boldsymbol{c}^*, \boldsymbol{p}^*)} < \tau < \frac{|\hat{c}_i - p_i^*|}{\mu(\hat{\boldsymbol{c}}, \boldsymbol{p}^*) + \frac{\|\boldsymbol{\delta}\|_1}{d}} - \frac{\max\left\{ \frac{\epsilon|v_j|}{\|\boldsymbol{v}\|_2}, \frac{\Delta KL(\boldsymbol{p}^*)v_j + \|\boldsymbol{v}\|_2\sqrt{-\Delta KL(\boldsymbol{p}^*)^2 + \epsilon^2\|\boldsymbol{v}\|_2^2}}{\|\boldsymbol{v}\|_2^2} \right\}}{\mu(\hat{\boldsymbol{c}}, \boldsymbol{p}^*) + \frac{\|\boldsymbol{\delta}\|_1}{d}} \tag{72}$$

To make $\tau$ feasible, we must have:

$$\frac{|c_j^* - p_j^*|}{\mu(\boldsymbol{c}^*, \boldsymbol{p}^*)} < \frac{|\hat{c}_i - p_i^*| - \max\left\{ \frac{\epsilon|v_j|}{\|\boldsymbol{v}\|_2}, \frac{\Delta KL(\boldsymbol{p}^*)v_j + \|\boldsymbol{v}\|_2\sqrt{-\Delta KL(\boldsymbol{p}^*)^2 + \epsilon^2\|\boldsymbol{v}\|_2^2}}{\|\boldsymbol{v}\|_2^2} \right\}}{\mu(\hat{\boldsymbol{c}}, \boldsymbol{p}^*) + \frac{\|\boldsymbol{\delta}\|_1}{d}} \tag{73}$$

Multiplying $\left( \mu(\hat{\boldsymbol{c}}, \boldsymbol{p}^*) + \frac{\|\boldsymbol{\delta}\|_1}{d} \right)$ on both sides yields:

$$\left( \mu(\hat{\boldsymbol{c}}, \boldsymbol{p}^*) + \frac{\|\boldsymbol{\delta}\|_1}{d} \right) \frac{|c_j^* - p_j^*|}{\mu(\boldsymbol{c}^*, \boldsymbol{p}^*)} < |\hat{c}_j - p_j^*| \tag{74}$$

$$- \max\left\{ \frac{\epsilon|v_j|}{\|\boldsymbol{v}\|_2}, \frac{\Delta KL(\boldsymbol{p}^*)v_j + \|\boldsymbol{v}\|_2\sqrt{-\Delta KL(\boldsymbol{p}^*)^2 + \epsilon^2\|\boldsymbol{v}\|_2^2}}{\|\boldsymbol{v}\|_2^2} \right\}. \tag{75}$$

By switching the LHS and RHS and then moving the $\max\left\{ \frac{\epsilon|v_j|}{\|\boldsymbol{v}\|_2}, \frac{\Delta KL(\boldsymbol{p}^*)v_j + \|\boldsymbol{v}\|_2\sqrt{-\Delta KL(\boldsymbol{p}^*)^2 + \epsilon^2\|\boldsymbol{v}\|_2^2}}{\|\boldsymbol{v}\|_2^2} \right\}$ term to the other side:

$$|\hat{c}_j - p_j^*| > \max\left\{ \frac{\epsilon|v_j|}{\|\boldsymbol{v}\|_2}, \frac{\Delta KL(\boldsymbol{p}^*)v_j + \|\boldsymbol{v}\|_2\sqrt{-\Delta KL(\boldsymbol{p}^*)^2 + \epsilon^2\|\boldsymbol{v}\|_2^2}}{\|\boldsymbol{v}\|_2^2} \right\} \tag{76}$$

$$+ \left( \mu(\hat{\boldsymbol{c}}, \boldsymbol{p}^*) + \frac{\|\boldsymbol{\delta}\|_1}{d} \right) \frac{|c_j^* - p_j^*|}{\mu(\boldsymbol{c}^*, \boldsymbol{p}^*)} \tag{77}$$

Then we can get a necessary condition on $|\hat{c}_j - c_j^*|$ via:

$$|\hat{c}_j - c_j^*| > \Big| |\hat{c}_j - p_j^*| - |c_j^* - p_j^*| \Big|$$

$$> \Big| max\Big\{ \frac{\epsilon |v_j|}{\|\boldsymbol{v}\|_2}, \frac{\Delta KL(\boldsymbol{p}^*)v_j + \|\boldsymbol{v}\|_2 \sqrt{-\Delta KL(\boldsymbol{p}^*)^2 + \epsilon^2 \|\boldsymbol{v}\|_2^2}}{\|\boldsymbol{v}\|_2^2} \Big\} \tag{78}$$

$$+ \Big( \mu(\hat{\boldsymbol{c}}, \boldsymbol{p}^*) + \frac{\|\boldsymbol{\delta}\|_1}{d} \Big) \frac{|c_j^* - p_j^*|}{\mu(\boldsymbol{c}^*, \boldsymbol{p}^*)} - |c_j^* - p_j^*| \Big|$$

$$= \Big| max\Big\{ \frac{\epsilon |v_j|}{\|\boldsymbol{v}\|_2}, \frac{\Delta KL(\boldsymbol{p}^*)v_j + \|\boldsymbol{v}\|_2 \sqrt{-\Delta KL(\boldsymbol{p}^*)^2 + \epsilon^2 \|\boldsymbol{v}\|_2^2}}{\|\boldsymbol{v}\|_2^2} \Big\} \tag{79}$$

$$+ \Big( \frac{\mu(\hat{\boldsymbol{c}}, \boldsymbol{p}^*)}{\mu(\boldsymbol{c}^*, \boldsymbol{p}^*)} + \frac{\|\boldsymbol{\delta}\|_1}{d \cdot \mu(\boldsymbol{c}^*, \boldsymbol{p}^*)} - 1 \Big) |c_j^* - p_j^*| \Big| \tag{80}$$

Therefore if $\frac{\epsilon |v_j|}{\|\boldsymbol{v}\|_2} > \frac{\Delta KL(\boldsymbol{p}^*)v_j + \|\boldsymbol{v}\|_2 \sqrt{-\Delta KL(\boldsymbol{p}^*)^2 + \epsilon^2 \|\boldsymbol{v}\|_2^2}}{\|\boldsymbol{v}\|_2^2}$, then we have:

$$|\hat{c}_j - c_j^*| > \Big| \frac{\epsilon |v_j|}{\|\boldsymbol{v}\|_2} + \Big( \frac{\mu(\hat{\boldsymbol{c}}, \boldsymbol{p}^*)}{\mu(\boldsymbol{c}^*, \boldsymbol{p}^*)} + \frac{\|\boldsymbol{\delta}\|_1}{d \cdot \mu(\boldsymbol{c}^*, \boldsymbol{p}^*)} - 1 \Big) |c_j^* - p_j^*| \Big|$$

$$\Downarrow \text{ Since each term is positive}$$

$$= \frac{\epsilon |v_j|}{\|\boldsymbol{v}\|_2} + \Big( \frac{\mu(\hat{\boldsymbol{c}}, \boldsymbol{p}^*)}{\mu(\boldsymbol{c}^*, \boldsymbol{p}^*)} + \frac{\|\boldsymbol{\delta}\|_1}{d \cdot \mu(\boldsymbol{c}^*, \boldsymbol{p}^*)} - 1 \Big) |c_j^* - p_j^*|$$

$$\Downarrow \text{ Since } \quad v_j = \frac{\hat{c}_j - c_j^*}{p_j^*}$$

$$= \frac{\epsilon |\hat{c}_j - c_j^*|}{\|\boldsymbol{v}\|_2 |p_j^*|} + \Big( \frac{\mu(\hat{\boldsymbol{c}}, \boldsymbol{p}^*)}{\mu(\boldsymbol{c}^*, \boldsymbol{p}^*)} + \frac{\|\boldsymbol{\delta}\|_1}{d \cdot \mu(\boldsymbol{c}^*, \boldsymbol{p}^*)} - 1 \Big) |c_j^* - p_j^*| \tag{81}$$

Moving all $|\hat{c}_j - c_j^*|$ to the left-hand-side:

$$|\hat{c}_j - c_j^*| > \frac{\Big( \frac{\mu(\hat{\boldsymbol{c}}, \boldsymbol{p}^*)}{\mu(\boldsymbol{c}^*, \boldsymbol{p}^*)} + \frac{\|\boldsymbol{\delta}\|_1}{d \cdot \mu(\boldsymbol{c}^*, \boldsymbol{p}^*)} - 1 \Big) |c_j^* - p_j^*|}{1 - \frac{\epsilon}{\|\boldsymbol{v}\| |p_j^*|}} \tag{82}$$

$$= \Big( \frac{\mu(\hat{\boldsymbol{c}}, \boldsymbol{p}^*)}{\mu(\boldsymbol{c}^*, \boldsymbol{p}^*)} + \frac{\|\boldsymbol{\delta}\|_1}{d \cdot \mu(\boldsymbol{c}^*, \boldsymbol{p}^*)} - 1 \Big) \frac{|c_j^* - p_j^*| \|\boldsymbol{v}\| |p_j^*|}{\|\boldsymbol{v}\| |p_j^*| - \epsilon} \tag{83}$$

On the other hand, if $\frac{\Delta KL(\boldsymbol{p}^*)v_j + \|\boldsymbol{v}\|_2 \sqrt{-\Delta KL(\boldsymbol{p}^*)^2 + \epsilon^2 \|\boldsymbol{v}\|_2^2}}{\|\boldsymbol{v}\|_2^2} > \frac{\epsilon |v_j|}{\|\boldsymbol{v}\|_2}$, then we have:

$$|\hat{c}_j - c_j^*| > \Big| \frac{\Delta KL(\boldsymbol{p}^*)v_j + \|\boldsymbol{v}\|_2 \sqrt{-\Delta KL(\boldsymbol{p}^*)^2 + \epsilon^2 \|\boldsymbol{v}\|_2^2}}{\|\boldsymbol{v}\|_2^2}$$

$$+ \Big( \frac{\mu(\hat{\boldsymbol{c}}, \boldsymbol{p}^*)}{\mu(\boldsymbol{c}^*, \boldsymbol{p}^*)} + \frac{\|\boldsymbol{\delta}\|_1}{d \cdot \mu(\boldsymbol{c}^*, \boldsymbol{p}^*)} - 1 \Big) |c_j^* - p_j^*| \Big|$$

$$= \frac{\Delta KL(\boldsymbol{p}^*)v_j + \|\boldsymbol{v}\|_2 \sqrt{-\Delta KL(\boldsymbol{p}^*)^2 + \epsilon^2 \|\boldsymbol{v}\|_2^2}}{\|\boldsymbol{v}\|_2^2}$$

$$+ \Big( \frac{\mu(\hat{\boldsymbol{c}}, \boldsymbol{p}^*)}{\mu(\boldsymbol{c}^*, \boldsymbol{p}^*)} + \frac{\|\boldsymbol{\delta}\|_1}{d \cdot \mu(\boldsymbol{c}^*, \boldsymbol{p}^*)} - 1 \Big) |c_j^* - p_j^*|$$

$$= \frac{\Delta KL(\boldsymbol{p}^*)v_j}{\|\boldsymbol{v}\|_2^2} + \frac{\|\boldsymbol{v}\|_2 \sqrt{-\Delta KL(\boldsymbol{p}^*)^2 + \epsilon^2 \|\boldsymbol{v}\|_2^2}}{\|\boldsymbol{v}\|_2^2}$$

$$+ \Big( \frac{\mu(\hat{\boldsymbol{c}}, \boldsymbol{p}^*)}{\mu(\boldsymbol{c}^*, \boldsymbol{p}^*)} + \frac{\|\boldsymbol{\delta}\|_1}{d \cdot \mu(\boldsymbol{c}^*, \boldsymbol{p}^*)} - 1 \Big) |c_j^* - p_j^*|$$

$$= \frac{\Delta KL(\boldsymbol{p}^*)(\hat{c}_j - c_j^*)}{\|\boldsymbol{v}\|_2^2 p_j^*} + \frac{\|\boldsymbol{v}\|_2 \sqrt{-\Delta KL(\boldsymbol{p}^*)^2 + \epsilon^2 \|\boldsymbol{v}\|_2^2}}{\|\boldsymbol{v}\|_2^2}$$

$$+ \Big( \frac{\mu(\hat{\boldsymbol{c}}, \boldsymbol{p}^*)}{\mu(\boldsymbol{c}^*, \boldsymbol{p}^*)} + \frac{\|\boldsymbol{\delta}\|_1}{d \cdot \mu(\boldsymbol{c}^*, \boldsymbol{p}^*)} - 1 \Big) |c_j^* - p_j^*|$$

$$= \frac{\Delta KL(\boldsymbol{p}^*)|\hat{c}_j - c_j^*|}{\|\boldsymbol{v}\|_2^2 p_j^* sgn(\hat{c}_j - c_j^*)} + \frac{\|\boldsymbol{v}\|_2 \sqrt{-\Delta KL(\boldsymbol{p}^*)^2 + \epsilon^2 \|\boldsymbol{v}\|_2^2}}{\|\boldsymbol{v}\|_2^2}$$

$$+ \Big( \frac{\mu(\hat{\boldsymbol{c}}, \boldsymbol{p}^*)}{\mu(\boldsymbol{c}^*, \boldsymbol{p}^*)} + \frac{\|\boldsymbol{\delta}\|_1}{d \cdot \mu(\boldsymbol{c}^*, \boldsymbol{p}^*)} - 1 \Big) |c_j^* - p_j^*|$$

Moving all term containing $|\hat{c}_j - c_j^*|$ to the left-hand-side:

$$|\hat{c}_j - c_j^*| > \frac{\frac{\|\boldsymbol{v}\|_2 \sqrt{-\Delta KL(\boldsymbol{p}^*)^2 + \epsilon^2 \|\boldsymbol{v}\|_2^2}}{\|\boldsymbol{v}\|_2^2} + \Big( \frac{\mu(\hat{\boldsymbol{c}}, \boldsymbol{p}^*)}{\mu(\boldsymbol{c}^*, \boldsymbol{p}^*)} + \frac{\|\boldsymbol{\delta}\|_1}{d \cdot \mu(\boldsymbol{c}^*, \boldsymbol{p}^*)} - 1 \Big) |c_j^* - p_j^*|}{1 - \frac{\Delta KL(\boldsymbol{p}^*)}{\|\boldsymbol{v}\|_2^2 p_j^* sgn(\hat{c}_j - c_j^*)}}$$

$$= \Bigg( \frac{\|\boldsymbol{v}\|_2 \sqrt{-\Delta KL(\boldsymbol{p}^*)^2 + \epsilon^2 \|\boldsymbol{v}\|_2^2}}{\|\boldsymbol{v}\|_2^2} + \Big( \frac{\mu(\hat{\boldsymbol{c}}, \boldsymbol{p}^*)}{\mu(\boldsymbol{c}^*, \boldsymbol{p}^*)} + \frac{\|\boldsymbol{\delta}\|_1}{d \cdot \mu(\boldsymbol{c}^*, \boldsymbol{p}^*)} - 1 \Big) |c_j^* - p_j^*| \Bigg)$$

$$\cdot \frac{\|\boldsymbol{v}\|_2^2 p_j^* sgn(\hat{c}_j - c_j^*)}{\|\boldsymbol{v}\|_2^2 p_j^* sgn(\hat{c}_j - c_j^*) - \Delta KL(\boldsymbol{p}^*)}$$

$$= \Bigg( \frac{\|\boldsymbol{v}\|_2 \sqrt{-\Delta KL(\boldsymbol{p}^*)^2 + \epsilon^2 \|\boldsymbol{v}\|_2^2}}{\|\boldsymbol{v}\|_2^2} + \Big( \frac{\mu(\hat{\boldsymbol{c}}, \boldsymbol{p}^*)}{\mu(\boldsymbol{c}^*, \boldsymbol{p}^*)} + \frac{\|\boldsymbol{\delta}\|_1}{d \cdot \mu(\boldsymbol{c}^*, \boldsymbol{p}^*)} - 1 \Big) |c_j^* - p_j^*| \Bigg)$$

$$\cdot \frac{\|\boldsymbol{v}\|_2^2 p_j^*}{\|\boldsymbol{v}\|_2^2 p_j^* - \Delta KL(\boldsymbol{p}^*) sgn(\hat{c}_j - c_j^*)}$$

$$= \frac{p_j^* \|\boldsymbol{v}\|_2 \sqrt{-\Delta KL(\boldsymbol{p}^*)^2 + \epsilon^2 \|\boldsymbol{v}\|_2^2}}{\|\boldsymbol{v}\|_2^2 p_j^* - \Delta KL(\boldsymbol{p}^*) sgn(\hat{c}_j - c_j^*)} \tag{84}$$

$$+ \Big( \frac{\mu(\hat{\boldsymbol{c}}, \boldsymbol{p}^*)}{\mu(\boldsymbol{c}^*, \boldsymbol{p}^*)} + \frac{\|\boldsymbol{\delta}\|_1}{d \cdot \mu(\boldsymbol{c}^*, \boldsymbol{p}^*)} - 1 \Big) \frac{|c_j^* - p_j^*| \|\boldsymbol{v}\|_2^2 p_j^*|}{\|\boldsymbol{v}\|_2^2 p_j^* - \Delta KL(\boldsymbol{p}^*) sgn(\hat{c}_j - c_j^*)} \tag{85}$$

Combining Equation 82, Equation 84, we can have:

$$|\hat{c}_j - c_j^*| > \max \Bigg\{ \Big( \frac{\mu(\hat{\boldsymbol{c}}, \boldsymbol{p}^*)}{\mu(\boldsymbol{c}^*, \boldsymbol{p}^*)} + \frac{\|\boldsymbol{\delta}\|_1}{d \cdot \mu(\boldsymbol{c}^*, \boldsymbol{p}^*)} - 1 \Big) \frac{|c_j^* - p_j^*| \|\boldsymbol{v}\|_2 p_j^*|}{\|\boldsymbol{v}\|_2 p_j^*| - \epsilon},$$

$$\frac{p_j^* \|\boldsymbol{v}\|_2 \sqrt{-\Delta KL(\boldsymbol{p}^*)^2 + \epsilon^2 \|\boldsymbol{v}\|_2^2}}{\|\boldsymbol{v}\|_2^2 p_j^* - \Delta KL(\boldsymbol{p}^*) sgn(\hat{c}_j - c_j^*)}$$

$$+ \Big( \frac{\mu(\hat{\boldsymbol{c}}, \boldsymbol{p}^*)}{\mu(\boldsymbol{c}^*, \boldsymbol{p}^*)} + \frac{\|\boldsymbol{\delta}\|_1}{d \cdot \mu(\boldsymbol{c}^*, \boldsymbol{p}^*)} - 1 \Big) \frac{|c_j^* - p_j^*| \|\boldsymbol{v}\|_2^2 p_j^*|}{\|\boldsymbol{v}\|_2^2 p_j^* - \Delta KL(\boldsymbol{p}^*) sgn(\hat{c}_j - c_j^*)} \Bigg\}$$

The Theorem holds by setting the threshold $\Gamma(\epsilon)$ to the right hand side of the inequality above. $\qquad \square$

