# OpenReview forum: "KoALA: KL–L0 Adversarial Detector via Label Agreement"
_ICLR.cc/2026/Conference — ICLR 2026 Conference Withdrawn Submission_

### Official Review · Reviewer_vJoS · 2025-10-29

**Soundness:** 3
**Presentation:** 3
**Contribution:** 2
**Rating:** 2
**Confidence:** 2

**Summary:**

The paper proposes a novel detection method against adversarial attacks which trains a detector using KL divergence and L0 based similarity indexes. The proposed method also provides formal proof of correctness with explicit conditions under which the proposed method would work.

**Strengths:**

1. Proposed defense is a plug-in module which is easy to implement without any training overhead.
2. The proposed method is grounded by theoretical foundations.

**Weaknesses:**

1. Evaluation Weakness: Attack evaluation is weak. Evaluation on PGD attack is not enough. The paper needs to test for gradient obfuscation (https://www.usenix.org/conference/usenixsecurity23/presentation/yue) using the test. The detector needs to be in the loop of attack with proper gradient flow.

2. Comparison to Competitive: What is the comparison with other adversarial attacks and defenses. No competitive method discussion in the result section. How does it compare against detection, training and other strategies.

3.  Lack of discussion on Overhead analysis: The detector has overhead (memory and time) at run-time. PGD training does not have overhead at run-time. The paper only discusses the second part which makes the comparison un-fair.

**Questions:**

Please refer to the three points in weaknesses.

---

### Official Review · Reviewer_84Bb · 2025-10-30

**Soundness:** 3
**Presentation:** 4
**Contribution:** 3
**Rating:** 8
**Confidence:** 4

**Summary:**

This paper introduces KoALA, a lightweight adversarial detector that flags inputs as adversarial when predictions from two complementary metrics—KL divergence and $L_0$-based similarity—disagree. The method requires no adversarial training or architectural changes, only a fine-tuning step on clean images. A formal proof guarantees detection under certain conditions. Experiments on ResNet/CIFAR-10 and CLIP/Tiny-ImageNet show strong performance.

**Strengths:**

- Originality: The idea of using the inherent disagreement between KL and $L_0$ metrics is novel and insightful.
- Theoretical Grounding: The formal guarantee is a major strength and differentiates it from many empirical detectors.
- Practicality: The method is simple, requires no adversarial training, and acts as a plug-in component.
- Clarity & Rigor: The presentation is top-tier, seamlessly blending theory and experiment.

**Weaknesses:**

- The performance on CLIP/Tiny-ImageNet, while good, is lower than on ResNet/CIFAR-10, as noted by the authors. The detector's effectiveness is contingent on the feature space structure achieved during fine-tuning.
- The related work section is adequate but could be slightly more comprehensive in contrasting with other "guarantee-seeking" lines of defense like randomized smoothing.

**Questions:**

1. How does KoALA's performance and overhead compare to a simple ensemble of models as a baseline for disagreement-based detection?
2. The fine-tuning step is "lightweight," but could you quantify the computational cost relative to standard adversarial training?
3. Could the theoretical framework be extended to other pairs of complementary metrics beyond KL and $L_0$?

---

### Official Review · Reviewer_DVcV · 2025-10-30

**Soundness:** 1
**Presentation:** 3
**Contribution:** 2
**Rating:** 2
**Confidence:** 5

**Summary:**

This paper introduces KoALA, a theoretically grounded adversarial example detector based on the disagreement between two complementary similarity measures: KL divergence and an $L_0$-based metric. The central idea is that adversarial perturbations can either be sparse, high-impact (affecting few dimensions with large magnitude) or dense, low-amplitude (affecting many dimensions slightly). By comparing predictions from KL and $L_0$–based prototype classifiers, KoALA detects adversarial examples when their outputs disagree. The authors provide a formal detection theorem proving that under several geometric assumptions, no perturbation can simultaneously deceive both metrics. The method is lightweight, requires only clean data for fine-tuning, and does not need adversarial training. Experiments on CIFAR-10 and Tiny-ImageNet demonstrate reasonable detection performance.

**Strengths:**

1. The dual-metric label agreement concept is elegant, interpretable, and easy to implement on top of existing models.
2. The paper provides theorems with clear assumptions, formal derivations, and mathematical reasoning connecting feature-space separability to detection correctness.
3. The structure is logical, the paper reads fluently, and the main idea is presented clearly. Figures help the reader understand the conceptual flow.

**Weaknesses:**

1. **Lack of motivating empirical validation for Section 3.1 assumptions.** The motivation relies on the assumption that adversarial perturbations can be categorized into sparse high-impact versus dense low-amplitude types under an energy-limited budget. However, the paper provides no empirical or statistical study validating this claim (e.g., perturbation sparsity distribution, histograms, or feature-level visualization). Since this assumption drives the entire method design, an ablation or exploratory study quantifying these two regimes would make the motivation much more convincing.
2. **No comparison with existing adversarial detectors.** The experiments include only internal ablations (KL-only, $L_0$-only, and cosine replacement) but no baselines against prior detectors such as Feature Squeezing, LID, Mahalanobis, or MagNet. Without these comparisons, it is difficult to assess the empirical competitiveness of KoALA. Including such baselines, even on one dataset, would significantly strengthen the paper.
3. **Inconsistency between theory ($\Vert \boldsymbol{\delta} \Vert_2$) and experiments ($\Vert \boldsymbol{\delta} \Vert_\infty$).** The theoretical results (Assumption A2) rely on an $L_2$-bounded perturbation, while the experiments use $L_\infty$ attacks (PGD, CW, AutoAttack). This inconsistency weakens the claimed theoretical guarantee. The authors should either (a) adjust A2 to match $L_\infty$ bounds, or (b) include experiments using $L_2$ attacks (e.g., PGD-$L_2$) to demonstrate that the theorem applies in practice.
4. **Missing ablation on the role of fine-tuning.** The paper claims the encoder fine-tuning step (Section 3.3) enforces agreement between KL and $L_0$ predictions on clean samples, but there is no ablation showing the performance without fine-tuning. This is crucial to demonstrate that fine-tuning is necessary and not an incidental enhancement. Adding a "no fine-tuning" variant would clarify how much of the performance improvement comes from the head versus encoder calibration.
5. **No hyperparameter sensitivity analysis.** Important parameters such as $\tau$, $\phi$, $\omega$ are fixed bot not analyzed. Reporting how performance changes with these values would help readers assess the method’s robustness and reproducibility.
6. **Issues and inaccuracies in theoretical derivations.** Several derivation steps in Appendix B contain mathematical inaccuracies:
    - B.1, after Eq. (10): In the substitution step after Eq. (10), the paper refers to “$\log(\hat{p}^\prime)$” which is undefined.
    - B.1, Eq. (11): The left-hand side is a vector, while the right-hand side contains vector and scalar terms. This is dimensionally inconsistent. Use elementwise Taylor expansion or rewrite with Hadamard operators $\odot,\oslash$.
    - B.1, Eq. (12)–(13) and below: The authors state, *“Since $\theta$, $p^{\ast}$, and $\delta$ all lie in the interval $[0,1]$ (thanks to Assumption A1) ... hence the remainder term increases as $\theta$ decreases, and its maximum is at $\theta = 0$”*.  This step is incorrect. Assumption A1 guarantees that the probability vectors $\boldsymbol{p}^{\ast}$ and the class prototypes $c$ are normalized and lie in $[0,1]$, but $\boldsymbol{\delta}$ represents the perturbation vector and may contain negative entries. The sign of each $\delta_i$ determines whether $|p_i^{\ast}+\theta\delta_i|$ increases or decreases with $\theta$, so monotonicity cannot be asserted globally. The inference that the cubic remainder term $R_3$ achieves its maximum at $\theta = 0$ is therefore unfounded. It is suggested to replace the monotonic argument with a uniform conservative bound based on Assumption A3 (bounded perturbation amplitude). Specifically, since $|\delta_i|\leq \frac{3}{2}|p_i^{\ast}|$, it follows that $|p_i^{\ast}+\theta\delta_i|\geq \frac{1}{2}|p_i|^{\ast}$ for any $\theta\in[0,1]$, and thus $R_3\leq\sum\limits_{i} \frac{8|\delta_i|^3}{3|p_i^\ast|^3}$. This maintains mathematical correctness without assuming $ \delta_i ≥ 0$.
    - B.1, before Eq. (14): The strict inequality “$< 0$” lacks justification; only “$\leq 0$” can be rigorously obtained.
    - B.3, Proposition 4: index ranges of subsets of $\mathbb{S}$ should use $\lbrace1,\dots,d\rbrace$ rather than $\lbrace1,\dots,m\rbrace$.
7. **Writing and presentation improvements.**
    - Section 3.3 (Fine-tuning): The process is unclear. The text should explicitly assume the availability of a batch of clean labeled data and describe how it is used for prototype computation and encoder tuning.
    - Add an overview paragraph before Section 3.1 to summarize the workflow of KoALA and Section 3.
    - Add a Conclusion section summarizing findings and limitations.
    - Reorganize figures/tables for better readability: Fig. 1 → top of p.2; Fig. 2 → top of p.4; Table 1 → top of p.7; Table 2 → top of p.8.
    - In Figure 1, the encoder should be drawn as a trapezoid (long left, short right) to visually indicate feature compression.
    - Table captions should be above the tables, not below.
    - Tables 1, 2, 4 should use a unified number of decimal places for consistency.
8. **Typos and notation/format issues.**
    - p.2, line 104: “of train … detectior” → “to train … detector”.
    - p.3, line 117: “comparision” → “comparison”.
    - p.3, lines 122 & 152: missing spaces before parentheses.
    - p.3, line 147 and later: use $\underset{k}{\arg\min}$ instead of $\arg\underset{k}{\min}$, and same for $\arg\max$.
    - p.5, line 220: write $\Vert \boldsymbol{δ} \Vert$ and specify the norm (should be $L_2$).
    - p.8, line 398: “Table 6” → “Table 2”.
    - Section titles 5, 6, 7: remove trailing colons.
    - Table 5 caption: missing space, “equation” → “Equation”.
    - Unify notation: use either $\mathrm{KL}(\cdot\Vert\cdot)$ or $\mathrm{KL}(\cdot,\cdot)$ consistently (prefer the former).
    - Standardize $L_0$ formatting: use $L_0$ consistently; similarly, use $\mathrm{KL}$ for upright font.
    - Vectors $\boldsymbol{c}, \boldsymbol{p}$ vs. $\mathbf{c}, \mathbf{p}$ – unify style.
    - In proofs, use upright fonts for $\log$, $\mathrm{diag}$, $\min$, $\max$, $\mathrm{sgn}$.
    - Bolden all the vectors (e.g., $\boldsymbol{\delta}$, $\boldsymbol{p}$) and keep consistency.

**Questions:**

See Weaknesses.

---

### Official Review · Reviewer_QTEW · 2025-11-04

**Soundness:** 3
**Presentation:** 3
**Contribution:** 3
**Rating:** 4
**Confidence:** 4

**Summary:**

This paper proposes an adversarial example detector called KOALA (KL–$L_0$ Adversarial detection via Label Agreement). The method compares the predicted class from two distance metrics: class with the minimal KL and $L_0$ between the input embedding and class prototypes. Experiment results on ResNet with CIFAR10 and Clip with Tiny-Imagenet are provided to demonstrate the efficacy of the method.

**Strengths:**

The paper proposes an adversarial detection method that requires no generation of adversarial examples and no architectural changes to the existing inference pipeline.

**Weaknesses:**

1. The evaluation uses only two datasets and a single architecture per dataset. I am not familiar with the literature on adversarial example detection, but this setting appears to be very limited.

2. There is no comparison with established detection methods.

3. The paper states that the detector requires only lightweight fine-tuning, but the cost is unclear, e.g. epochs, hyper-param search, etc.

4. Can we also generate adversarial examples using the fine-tuned detector?

5. Assumption A3 is described as mild/practical. Can you cite prior work using the same assumption and provide empirical evidence that "extremely large, coordinate-wise perturbations are rarely effective or imperceptible."?

6. It is unclear how we can observe "we observe that energy-bounded attacks manifest either as (i) dense, low-amplitude shifts across many coordinates or (ii) sparse, high-impact shifts on few coordinates." from Figure 1. Similar issue with the statement on line 156-157.

**Questions:**

Suggestions:
1. Mention the normalization of the feature vectors earlier (e.g. around Ln162) to avoid confusion.
2. Clarifying $\mathbf{c}_i$ and  $c_i$ explicitly.
3. The statement on Ln435 "indicating that the fine-tuning process does not degrade the model’s core classification ability." should be corrected. There is a consistent but minor drop in accuracy across all finetuning.

Questions:
1. It appears that there are a lot of hyper parameters that needs to be optimized. For example, $w_{L_0}$ $w_{KL}$, $\tau$, $\phi$. Can they be easily identified in practice or requires exhaustive searches? Also, how sensitive is the result to those hyper parameters?
2. The baseline model row in Table 3 is the ResNet18 with the original classifier head? Why is the accuracy under attacks so high?
3. How many epochs are the models fine-tuned for?

---

### Note · Authors · 2025-11-24

I have read and agree with the venue's withdrawal policy on behalf of myself and my co-authors.